# A large modulation of electron-phonon coupling and an emergent superconducting dome in doped strong ferroelectrics

Jiaji Ma[1], Ruihan Yang[1] & Hanghui Chen [1,2✉]

We use first-principles methods to study doped strong ferroelectrics (taking $BaTiO_3$ as a prototype). Here, we find a strong coupling between itinerant electrons and soft polar phonons in doped $BaTiO_3$, contrary to Anderson/Blount's weakly coupled electron mechanism for "ferroelectric-like metals". As a consequence, across a polar-to-centrosymmetric phase transition in doped $BaTiO_3$, the total electron-phonon coupling is increased to about 0.6 around the critical concentration, which is sufficient to induce phonon-mediated superconductivity of about 2 K. Lowering the crystal symmetry of doped $BaTiO_3$ by imposing epitaxial strain can further increase the superconducting temperature via a sizable coupling between itinerant electrons and acoustic phonons. Our work demonstrates a viable approach to modulating electron-phonon coupling and inducing phonon-mediated superconductivity in doped strong ferroelectrics and potentially in polar metals. Our results also show that the weakly coupled electron mechanism for "ferroelectric-like metals" is not necessarily present in doped strong ferroelectrics.

---

[1] NYU-ECNU Institute of Physics, NYU Shanghai, Shanghai, China. [2] Department of Physics, New York University, New York, NY, USA.
✉email: hanghui.chen@nyu.edu

Electron-phonon coupling plays an important role in a variety of physical phenomena in solids[1]. In metals and doped semiconductors, low-energy electronic excitations are strongly modified by the coupling of itinerant electrons to lattice vibrations, which influences their transport and thermodynamic properties[2]. Furthermore, electron-phonon coupling provides an attractive electron-electron interaction, which leads to conventional (i.e., phonon-mediated) superconductivity in many metals[3]. Recent studies on hydrogen-rich materials show that when their electron-phonon coupling is strong enough, the transition temperature of conventional superconductors can reach as high as 260 K at 180–200 GPa[4–6]. One general way to increase the electron-phonon coupling of solids is to find a particular phonon to which itinerant electrons are strongly coupled and whose softening (i.e., the phonon frequency approaches zero) across a structural phase transition may consequently increase the total electron-phonon coupling[7]. However, identifying a strong coupling between a soft phonon and itinerant electrons in real materials is no easy task, which relies on material details. On the other hand, the superconductivity in doped $SrTiO_3$ has drawn great interests from both theorists[8–15] and experimentalists[16–24]. One beautiful experiment is $Sr_{1-x}Ca_xTiO_{3-\delta}$ in which Ca doping leads to a weak ferroelectric distortion in $SrTiO_3$ and oxygen vacancies provide itinerant electrons[20,25,26]. Increasing the carrier concentration in $Sr_{1-x}Ca_xTiO_{3-\delta}$ induces a polar-to-centrosymmetric phase transition and a superconducting "dome" emerges around the critical concentration. The nature of the superconductivity in doped $SrTiO_3$ is highly debatable[8–21,27], because the superconductivity in doped $SrTiO_3$ can persist to very low carrier density[11,12,28], which seriously challenges the standard phonon pairing mechanism[29]. It is not clear why superconductivity in doped $SrTiO_3$ vanishes above a critical concentration in spite of an increasing density of states at the Fermi level[30]. Attention has been paid to recent proposals on soft polar phonons, but the coupling details and strength are controversial[10,11,15,27,31]. Furthermore, according to Anderson and Blount's original proposal that inversion symmetry breaking by collective polar displacements in metals relies on the weak coupling between itinerant electrons and soft phonons responsible for inversion symmetry breaking[32–34], it is not obvious that across the polar-to-centrosymmetric phase transition the soft polar phonons can be coupled to itinerant electrons in $Sr_{1-x}Ca_xTiO_{3-\delta}$, or more generally in doped ferroelectrics and polar metals[11,15,31,35].

Motivated by the above experiments and theories, we use first-principle methods with no adjustable parameters to demonstrate a large modulation of electron-phonon coupling in doped strong ferroelectrics by utilizing soft polar phonons. We study $BaTiO_3$ as a prototype, because (1) previous studies found that in $n$-doped $BaTiO_3$, increasing the carrier density gradually reduces its polar distortions and induces a continuous polar-to-centrosymmetric phase transition[36,37]; and (2) the critical concentration for the phase transition is about $10^{21}/cm^3$, which is high enough so that the electron-phonon coupling can be directly calculated within the Migdal's approximation (in contrast, in doped $SrTiO_3$, superconductivity emerges at a much lower carrier concentration $10^{17}$–$10^{20}/cm^3$ so that its Debye frequency is comparable to or even higher than the Fermi energy $\hbar\omega_D/\epsilon_F \sim 1-10^{2}$[38], which invalidates the Migdal's approximation and Eliashberg equation)[29]. The key result from our calculation is that, contrary to Anderson/Blount's argument for "ferroelectric-like metals"[32–34], we find that the phonon bands associated with the soft polar optical phonons are strongly coupled to itinerant electrons across the polar-to-centrosymmetric phase transition in doped $BaTiO_3$. As a consequence, the total electron-phonon coupling of doped $BaTiO_3$ can be substantially modulated via carrier density and in particular is increased to about 0.6 around the critical concentration. Eliashberg equation calculations find that such an electron-phonon coupling is

sufficiently large to induce phonon-mediated superconductivity of about 2K. In addition, we find that close to the critical concentration, lowering the crystal symmetry of doped $BaTiO_3$ by imposing epitaxial strain further increases the superconducting temperature via a sizable coupling between itinerant electrons and acoustic phonon bands.

While ferroelectricity and superconductivity have little in common, our work demonstrates an experimentally viable approach to modulating electron-phonon coupling and inducing phonon-mediated superconductivity in doped strong ferroelectrics and potentially in polar metals[32,39]. Our results show that the weakly coupled electron mechanism in "ferroelectric-like metals" is not necessarily present in doped strong ferroelectrics and as a consequence, the soft polar phonons can be utilized to induce phonon-mediated superconductivity across a structural phase transition.

## Results

**Structural phase transition induced by electron doping**. In this study, electron doping in $BaTiO_3$ is achieved by adding extra electrons to the system with the same amount of uniform positive charges in the background. For benchmarking, our calculation of the undoped tetragonal $BaTiO_3$ gives the lattice constant $a = 3.930$ Å and $c/a = 1.012$, polarization $P = 0.26$ C/m$^2$, and Ti-O and Ba-O relative displacements of 0.105 Å and 0.083 Å, respectively, consistent with the previous calculations[40–42]. We note that upon electron doping, $BaTiO_3$ becomes metallic and its polarization is ill-defined[43]. Therefore, we focus on analyzing ionic polar displacements and $c/a$ ratio to identify the critical concentration[36].

We test four different crystal structures of $BaTiO_3$ with electron doping: the rhombohedral structure (space group $R3m$ with Ti displaced along $\langle 111 \rangle$ direction), the orthorhombic structure (space group $Amm2$ with Ti displaced along $\langle 011 \rangle$ direction), the tetragonal structure (space group $P4mm$ with Ti displaced along $\langle 001 \rangle$ direction) and the cubic structure (space group $Pm\bar{3}m$ with Ti at the center of oxygen octahedron). Figure 1a shows that as electron doping concentration $n$ increases from 0 to $0.15e$/f.u., $BaTiO_3$ transitions from the rhombohedral structure to the tetragonal structure, and finally to the cubic structure. The critical concentration is such that the crystal structure of doped $BaTiO_3$ continuously changes from tetragonal to cubic (see Supplementary Note 6). While the structural transition from tetragonal to cubic is continuous, the transition from rhombohedral to tetragonal is first-order and thus does not show phonon softening (see Supplementary Note 7). Furthermore the low electron concentration in the rhombohedral structure invalidates Migdal's theorem and electron-phonon coupling cannot be calculated within Migdal's approximation (see Supplementary Note 5).

Figure 1b shows $c/a$ ratio and Ti-O cation displacements $\delta$ as a function of the concentration $n$ in the range of 0.06–0.14$e$/f.u. It is evident that the critical concentration $n_c$ of doped $BaTiO_3$ is $0.10e$/f.u. (about $1.6 \times 10^{21}$ cm$^{-3}$), at which the polar displacement $\delta$ is just completely suppressed and the $c/a$ ratio is reduced to unity. This result is consistent with the previous theoretical studies[36,37]. Experimentally, in metallic oxygen-deficient $BaTiO_{3-\delta}$, the low-symmetry polar structure can be retained up to an electron concentration of $1.9 \times 10^{21}$ cm$^{-3}$ (close to the theoretical result)[44,45]. However, weak localization and/or phase separation may exist in oxygen-deficient $BaTiO_3$, depending on sample quality[44,46].

**Electronic structure and phonon properties**. Figure 2a shows the electronic structure of doped $BaTiO_3$ in the tetragonal

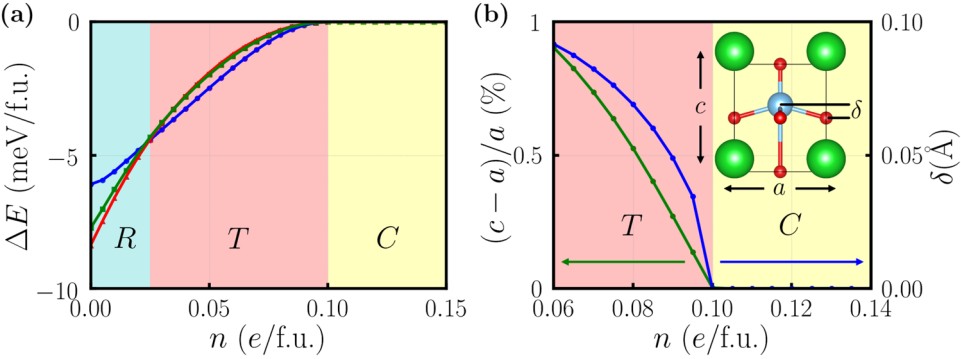

**Fig. 1 Structural phase transition of BaTiO$_3$ induced by electron doping. a** Total energies of $n$-doped BaTiO$_3$ in different crystal structures: the rhombohedral structure ($R$, red line), the orthorhombic structure ($O$, green line), the tetragonal structure ($T$, blue line) and the cubic structure ($C$, setting as the zero point at each electron doping concentration $n$). Upon electron doping, the ground state structure of BaTiO$_3$ changes from $R$ to $T$, finally to $C$. **b** The $c/a$ ratio and Ti-O cation displacement $\delta$ of $n$-doped BaTiO$_3$. $T$ means the tetragonal structure and $C$ means the cubic structure. The inset shows the tetragonal structure of doped BaTiO$_3$ where $c$ is the long cell axis and $a$ is the short cell axis. $\delta$ is the displacement of the Ti atom with respect to the O atom layer along the $c$ axis.

structure at a representative concentration ($n = 0.09e$/f.u., close to the critical value). Undoped BaTiO$_3$ is a wide gap insulator. Electron doping moves the Fermi level slightly above the conduction band edge of the three Ti $t_{2g}$ orbitals and thus a Fermi surface is formed. We use three Wannier functions to reproduce the Ti $t_{2g}$ bands, upon which electron-phonon coupling is calculated. Figure 2b shows the phonon spectrum of doped BaTiO$_3$ in the tetragonal structure at $0.09e$/f.u. concentration. We are particularly interested in the zone-center (Γ-point) polar optical phonons, which are highlighted by the green dots in Fig. 2b. The vibrational modes of those polar phonons are explicitly shown in Fig. 2c. In the tetragonal structure of BaTiO$_3$, the two polar phonons with the ion displacements along $x$ and $y$ directions ($\omega_x$ and $\omega_y$) are degenerate, while the third polar phonon with the ion displacements along $z$ direction ($\omega_z$) has higher frequency. Figure 2d shows that electron doping softens the zone-center polar phonons of BaTiO$_3$ in the tetragonal structure until it reaches the critical concentration where the three polar phonon frequencies become zero. With further electron doping, the polar phonon frequencies of BaTiO$_3$ increase in the cubic structure (see Supplementary Note 13 for a discussion about doping's effect on polar phonon behavior).

**Electron-phonon coupling and phonon-mediated superconductivity.** The continuous polar-to-centrosymmetric phase transition in doped BaTiO$_3$ is similar to the one in "ferroelectric-like metals" proposed by Anderson and Blount[32]. They first argued, later recast by Puggioni and Rondinelli[33,34], that inversion symmetry breaking by collective polar displacements in a metal relies on a weak coupling between itinerant electrons and soft phonons responsible for removing inversion symmetry. According to this argument, one would expect that across the polar-to-centrosymmetric phase transition, the soft polar phonons are not strongly coupled to itinerant electrons in doped BaTiO$_3$. In order to quantify the strength of electron-phonon coupling and make quantitative comparison, we introduce the mode-resolved electron-phonon coupling $\lambda_{\mathbf{q}\nu}$ and around-zone-center branch-resolved electron-phonon coupling $\lambda_\nu$:

$$\lambda_{\mathbf{q}\nu} = \frac{1}{\pi N_F}\frac{\mathrm{Im}\Pi_{\mathbf{q}\nu}}{\omega_{\mathbf{q}\nu}^2} \quad \text{and} \quad \lambda_\nu = \frac{\int_{|\mathbf{q}|<q_c} d\mathbf{q}\lambda_{\mathbf{q}\nu}}{\int_{|\mathbf{q}|<q_c} d\mathbf{q}} \quad (1)$$

where $\mathrm{Im}\Pi_{\mathbf{q}\nu}$ is the imaginary part of electron-phonon self-energy, $\omega_{\mathbf{q}\nu}$ is the phonon frequency, $N_F$ is the density of states at the Fermi level and $q_c$ is a small phonon momentum. The reason

we define $\lambda_\nu$ within $|\mathbf{q}| < q_c$ is because: (1) exactly at the zone-center Γ point, the acoustic phonon frequency is zero and thus the contribution from the acoustic mode is ill-defined at Γ point; (2) when $q_c$ is sufficiently small, there are no phonon band crossings within $|\mathbf{q}| < q_c$ and hence each branch $\nu$ can be assigned to a well-defined phonon mode (for a general $\mathbf{q}$ point, it is not trivial to distinguish which phonon band corresponds to polar modes and which to other optical modes). We choose $q_c = 0.05\frac{\pi}{a}$ where $a$ is the lattice constant (the qualitative conclusions do not depend on the choice of $q_c$, as long as no phonon band crossings occur within $|\mathbf{q}| < q_c$).

Figure 3a, b show the imaginary part of electron-phonon self-energy $\mathrm{Im}\Pi_{\mathbf{q}\nu}$ for each phonon mode $\mathbf{q}\nu$ of doped BaTiO$_3$ along a high-symmetry path (panel a corresponds to $0.09e$/f.u. doping in a tetragonal structure and panel b corresponds to $0.11e$/f.u. doping in a cubic structure). Since within the double delta approximation $\mathrm{Im}\Sigma_{\mathbf{q}\nu}$ is positive definite (see Supplementary Note 2), the point size in panels a and b is chosen to be proportional to the value of $\mathrm{Im}\Pi_{\mathbf{q}\nu}$. Our calculations find that, contrary to Anderson/Blount's weak coupled electron mechanism[32], the phonon bands associated with the zone-center polar phonons have the strongest coupling to itinerant electrons, while the couplings of other phonon bands are weaker. Specifically, in the case of $0.09e$/f.u. doping:

$$\lambda_{\mathrm{acoustic}} = \sum_{\nu=1-3}\lambda_\nu = 3.83 \quad (2)$$

$$\lambda_{\mathrm{polar}} = \sum_{\nu=4-6}\lambda_\nu = 10.92$$

$$\lambda_{\mathrm{others}} = \sum_{\nu=7-15}\lambda_\nu = 0.53$$

and in the case of $0.11e$/f.u. doping:

$$\lambda_{\mathrm{acoustic}} = \sum_{\nu=1-3}\lambda_\nu = 0.27 \quad (3)$$

$$\lambda_{\mathrm{polar}} = \sum_{\nu=4-6}\lambda_\nu = 5.58$$

$$\lambda_{\mathrm{others}} = \sum_{\nu=7-15}\lambda_\nu = 0.11$$

In both cases, $\lambda_{\mathrm{polar}}$ is larger than $\lambda_{\mathrm{acoustic}}$ and $\lambda_{\mathrm{others}}$. An intuitive picture for the strong coupling is that in doped BaTiO$_3$, the soft polar phonons involve the cation displacements of Ti and O atoms, and in the meantime itinerant electrons derive from Ti-$d$ states which hybridize with O-$p$ states (see Supplementary

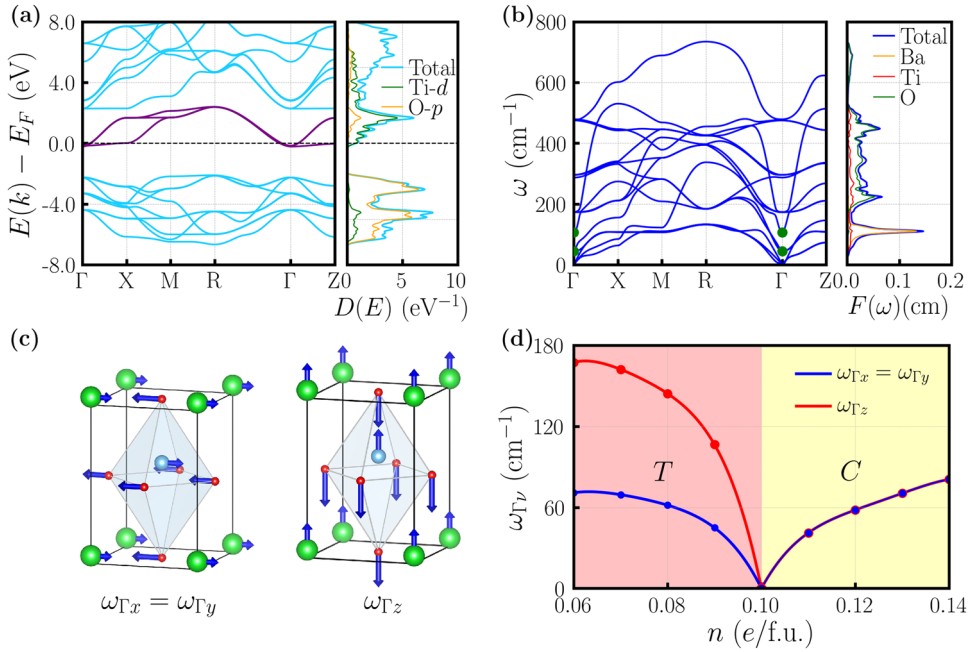

**Fig. 2 Electronic structure and phonon properties of doped BaTiO₃.** **a** Electronic band structure and density of states of doped BaTiO₃ in the tetragonal structure at 0.09e/f.u. concentration. In the electronic band structure, the three purple bands are generated by three maximally localized Wannier functions that exactly reproduce the original Ti $t_{2g}$ bands. In the electronic density of states, the blue, green and orange curves correspond to total, Ti-$d$ projected and O-$p$ projected partial densities of states, respectively. **b** Phonon band structure and phonon density of states of doped BaTiO₃ in the tetragonal structure at 0.09e/f.u. concentration. In the phonon band structure, the green dots highlight the zone-center polar optical phonons. In the phonon density of states, the blue, orange, red and green curves correspond to total, Ba-projected, Ti-projected and O-projected partial densities of states, respectively. **c** Vibration modes of the zone-center polar optical phonons of doped BaTiO₃ in the tetragonal structure at 0.09e/f.u. concentration. The left panel shows that the atoms of BaTiO₃ are vibrating along the short $a$ axis (either $x$-axis or $y$-axis, degenerate due to the tetragonal symmetry). The right panel shows that the atoms of BaTiO₃ are vibrating along the long $c$ axis ($z$-axis). **d** The frequencies of the three zone-center polar phonons of doped BaTiO₃ as a function of electron concentration $n$. The critical concentration is at 0.1e/f.u. where the polar phonon frequencies are reduced to zero.

Note 14 for an alternative demonstration of this strong coupling, and Supplementary Note 12 for a discussion about doping's effect on this $p$–$d$ hybridization). This is in contrast to the textbook example of polar metals LiOsO₃ where the soft polar phonons involve Li displacements while the metallicity derives from Os and O orbitals[33]. More quantitatively, we find $\lambda_{polar} = 0.50$ for LiOsO₃, which is substantially smaller than $\lambda_{polar}$ of about 5–10 for doped BaTiO₃. In short, because the itinerant electrons and polar phonons are associated with the same atoms in doped BaTiO₃, the coupling is strong, while in LiOsO₃ the itinerant electrons and polar phonons involve different atoms and thus the coupling is weak. As a consequence of the strong interaction between the polar phonons and itinerant electrons, we expect that the total electron-phonon coupling of doped BaTiO₃ can be increased by softening the polar phonons across the structural phase transition.

Figure 3c shows the total electron-phonon spectral function $\alpha^2 F(\omega)$ and accumulative electron-phonon coupling $\lambda(\omega)$ of doped BaTiO₃ at 0.09e/f.u. and 0.11e/f.u. concentrations. $\alpha^2 F(\omega)$ is defined as:

$$\alpha^2 F(\omega) = \frac{1}{2}\sum_\nu \int \frac{d\mathbf{q}}{\Omega_{BZ}} \omega_{\mathbf{q}\nu}\lambda_{\mathbf{q}\nu}\delta(\omega - \omega_{\mathbf{q}\nu}) \quad (4)$$

where $\Omega_{BZ}$ is the volume of phonon Brillouin zone. With $\alpha^2 F(\omega)$, it is easy to calculate the accumulative electron-phonon coupling $\lambda(\omega)$:

$$\lambda(\omega) = 2\int_0^\omega \frac{\alpha^2 F(\nu)}{\nu}\,d\nu \quad (5)$$

The total electron-phonon coupling $\lambda$ is obtained by taking the upper bound $\omega$ to $\infty$ in Eq. (5). The green shades are $\alpha^2 F(\omega)$ and

the dashed lines are the corresponding cumulative electron-phonon coupling. The total electron-phonon coupling $\lambda$ of doped BaTiO₃ in the tetragonal structure at 0.09e/f.u. concentration is 0.61, while that in the cubic structure at 0.11e/f.u. concentration is 0.50. Both $\lambda$ are sufficiently large to induce phonon-mediated superconductivity with measurable transition temperature. Figure 3d shows the total electron-phonon coupling $\lambda$ of doped BaTiO₃ for a range of electron concentrations (exactly at the critical concentration, we find some numerical instabilities and divergence in the electron-phonon calculations, rendering the result unreliable). An increase of $\lambda$ around the critical concentration is evident, consistent with the strong coupling between the soft polar phonons and itinerant electrons in doped BaTiO₃.

Based on the electron-phonon spectrum $\alpha^2 F(\omega)$, we use a three-orbital Eliashberg equation (see Supplementary Note 3) to calculate the superconducting gap $\Delta(T)$ and estimate the superconducting transition temperature $T_c$ as a function of electron concentration. Because the three Ti $t_{2g}$ orbitals become identical at the critical concentration, when solving the three-orbital Eliashberg equation, we set Morel-Anderson pseudopotential $\mu_{ij}^*$ to be 0.1 for each orbital pair (i.e., $i, j = 1, 2, 3$)[47]. Figure 3e shows the superconducting gap $\Delta(T)$ of doped BaTiO₃ as a function of temperature $T$ at two representative concentrations (0.09e/f.u. in the tetragonal structure and 0.11e/f.u. in the cubic structure). Since both concentrations are close to the critical value, the three Ti $t_{2g}$ orbitals are almost degenerate in doped BaTiO₃. For clarification, we show the superconducting gap of one orbital for each concentration. From the Eliashberg equation, we find that at 0.09e/f.u. concentration, $\Delta(T = 0) = 0.27$ meV and $T_c = 1.75$ K; and at

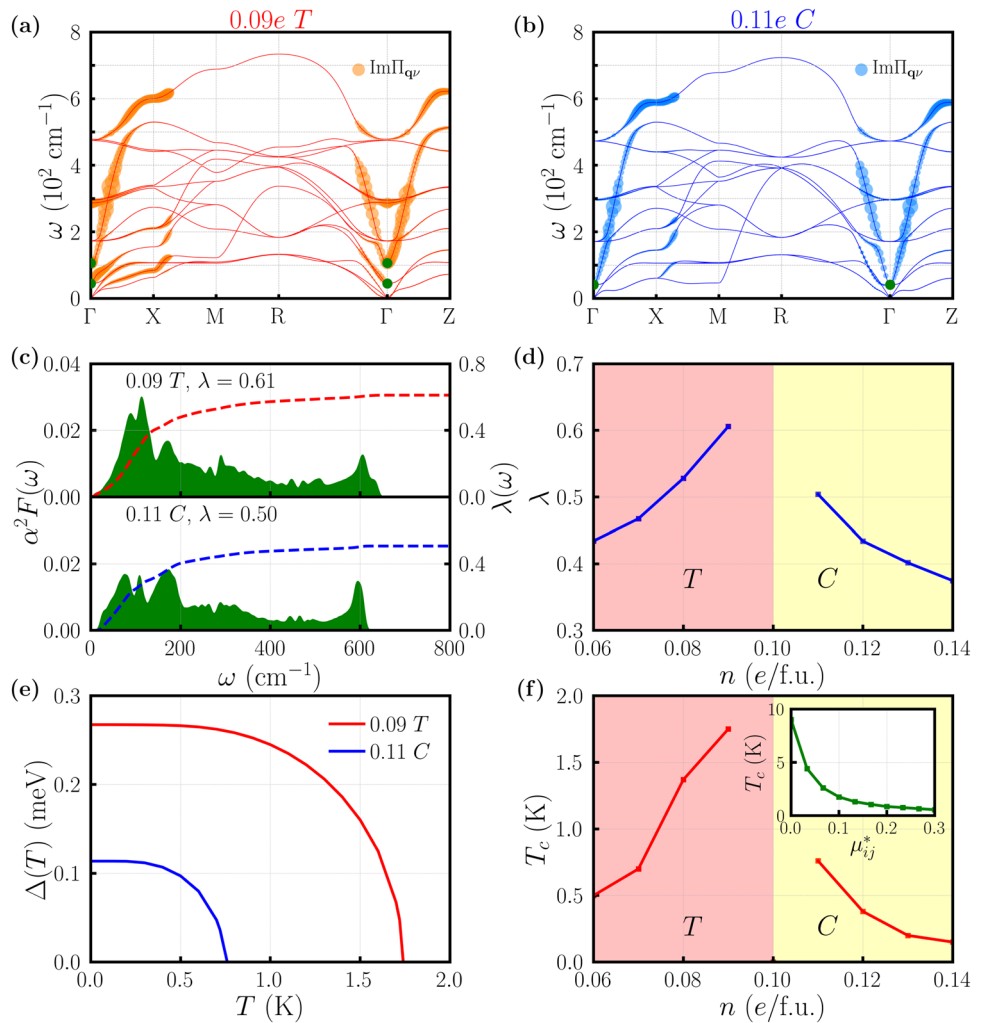

**Fig. 3 Electron-phonon properties and phonon-mediated superconductivity in doped BaTiO₃. a** The imaginary part of the electron-phonon self-energy ImΠ$_{\mathbf{q}\nu}$ for each phonon mode of doped BaTiO₃ at 0.09e/f.u. concentration (tetragonal structure $T$). The point size is proportional to ImΠ$_{\mathbf{q}\nu}$. The largest point corresponds to ImΠ$_{\mathbf{q}\nu}$ = 4.6 meV. The green dots highlight the zone-center polar optical phonons. **b** The imaginary part of the electron-phonon self-energy ImΠ$_{\mathbf{q}\nu}$ for each phonon mode of doped BaTiO₃ at 0.11e/f.u. concentration (cubic structure $C$). The point size is proportional to ImΠ$_{\mathbf{q}\nu}$. The largest point corresponds to ImΠ$_{\mathbf{q}\nu}$ = 3.2 meV. The green dots highlight the zone-center polar optical phonons. **c** Electron-phonon spectral function $\alpha^2F(\omega)$ and accumulative electron-phonon coupling $\lambda(\omega)$ of doped BaTiO₃ at 0.09e/f.u. and 0.11e/f.u. concentration. The total electron-phonon coupling $\lambda$ is 0.61 for the former and 0.50 for the latter. **d** Total electron-phonon coupling $\lambda$ of doped BaTiO₃ as a function of electron concentration $n$. **e** Superconducting gap $\Delta$ of doped BaTiO₃ as a function of temperature $T$ at 0.09 e/f.u. concentration (red) and at 0.11 e/f.u. concentration (blue), calculated by the three-orbital Eliashberg equation. The Morel-Anderson pseudopotential $\mu^*_{ij}$ = 0.1 is used for each orbital pair. **f** Superconducting transition temperature $T_c$ of doped BaTiO₃ calculated by the Eliashberg equation as a function of electron concentration $n$. The inset shows $T_c$ of BaTiO₃ in the tetragonal structure at 0.09e/f.u. concentration as a function of Morel-Anderson pseudopotential $\mu^*_{ij}$.

0.11e/f.u. concentration, $\Delta(T = 0)$ = 0.11 meV and $T_c$ = 0.76 K. Thus $\Delta(T=0)/(k_BT_c)$ = 1.79 at 0.09e/f.u. concentration and 1.68 at 0.11e/f.u. concentration, both close to the BCS prediction of 1.77. Figure 3f shows the estimated superconducting transition temperature $T_c$ of doped BaTiO₃ for a range of electron concentrations. $T_c$ notably exhibits a dome-like feature as a function of electron concentration. The origin of the superconducting "dome" is that the electron-phonon coupling of doped BaTiO₃ is increased by the softened polar phonons around the critical concentration. When the electron concentration is away from the critical value, the polar phonons are "hardened" (i.e., phonon frequency increases) and the electron-phonon coupling of doped BaTiO₃ decreases. We note that the estimated $T_c$ strongly depends on $\mu^*_{ij}$. Therefore in the inset of Fig. 3f, we study doped BaTiO₃ at a representative concentration (0.09e/f.u.) and calculate its superconducting transition temperature $T_c$ as a function of $\mu^*_{ij}$.

As $\mu^*_{ij}$ changes from 0 to 0.3, the estimated $T_c$ decreases from 9.3 K to 0.4 K, the lowest of which (0.4 K) is still measurable in experiment[28]. We make two comments here: (1) The superconducting transition temperature is only an estimation due to the uncertainty of Morel-Anderson pseudopotential $\mu^*_{ij}$ and other technical details. But the picture of an increased electron-phonon coupling around the structural phase transition in doped BaTiO₃ is robust. (2) Experimentally in Sr$_{1-x}$Ca$_x$TiO₃, the optimal doping for superconductivity is larger than the "ferroelectric" critical concentration[20], while in our calculations of doped BaTiO₃, the two critical concentrations (one for optimal superconducting $T_c$ and the other for suppressing polar displacements) just coincide due to polar phonon softening and an increased electron-phonon coupling. Comparison of these two materials implies that the microscopic mechanism for superconductivity in doped SrTiO₃ is probably not purely phonon-mediated.

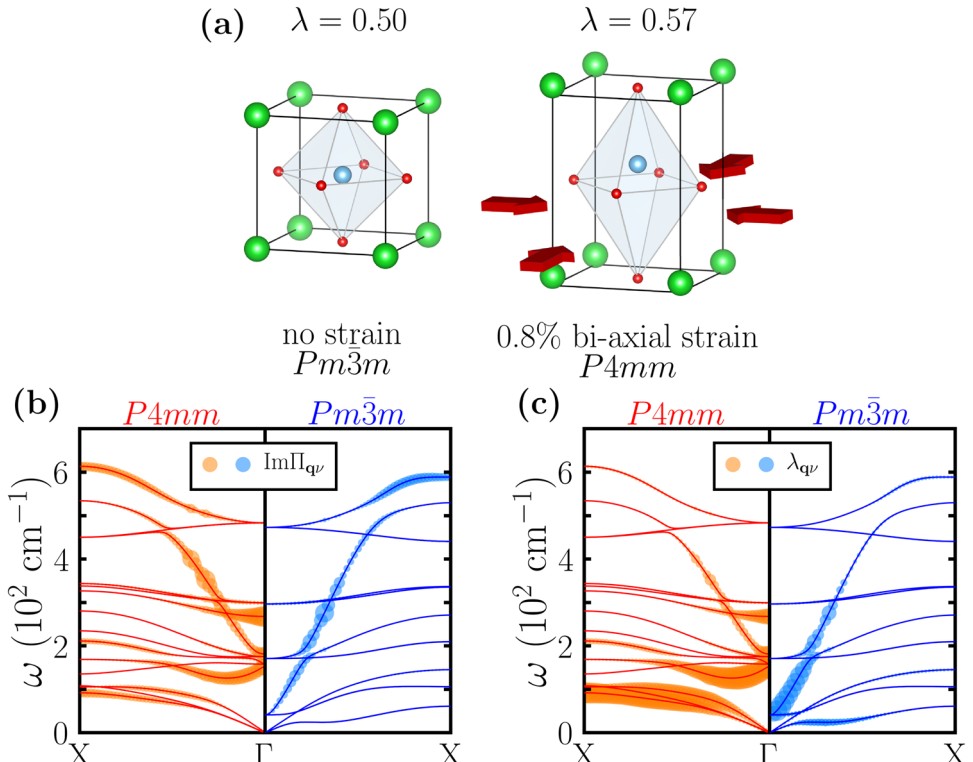

**Fig. 4 Comparison of doped BaTiO₃ between the cubic structure and the strain-induced tetragonal structure. a** Doped BaTiO₃ at 0.11e/f.u. concentration. Left is the cubic crystal structure of BaTiO₃ with no strain (space group $Pm\bar{3}m$) and right is the polar tetragonal crystal structure of BaTiO₃ under 0.8% bi-axial strain (space group $P4mm$). **b** The imaginary part of the electron-phonon self-energy $\mathrm{Im}\Pi_{\mathbf{q}\nu}$ for each phonon mode of doped BaTiO₃, at 0.11e/f.u. in the $P4mm$ structure (red, left) and in the $Pm\bar{3}m$ structure (blue, right). The point size is proportional to $\mathrm{Im}\Pi_{\mathbf{q}\nu}$. The largest point corresponds to $\mathrm{Im}\Pi_{\mathbf{q}\nu} = 3.8$ meV. **c** Mode-resolved electron-phonon coupling $\lambda_{\mathbf{q}\nu}$ for each phonon mode of doped BaTiO₃, at 0.11e/f.u. in the $P4mm$ structure (red, left) and in the $Pm\bar{3}m$ structure (blue, right). The point size is proportional to $\lambda_{\mathbf{q}\nu}$. The largest point corresponds to $\lambda_{\mathbf{q}\nu} = 5.1$.

**Crystal symmetry and acoustic phonons**. We note that in Figure 3a, b, in addition to the large $\mathrm{Im}\Pi_{\mathbf{q}\nu}$ in the polar optical phonon bands, there is also sizable $\mathrm{Im}\Pi_{\mathbf{q}\nu}$ in the acoustic phonon bands (from Γ to X) in the tetragonal structure at 0.09e/f.u. concentration. Since the mode-resolved electron-phonon coupling $\lambda_{\mathbf{q}\nu} \propto \mathrm{Im}\Pi_{\mathbf{q}\nu}/\omega_{\mathbf{q}\nu}^2$, the small frequency of acoustic phonons can lead to a substantial $\lambda_{\mathbf{q}\nu}$, given a sizable $\mathrm{Im}\Pi_{\mathbf{q}\nu}$. However, in the cubic structure at 0.11e/f.u. concentration, $\mathrm{Im}\Pi_{\mathbf{q}\nu}$ in the acoustic phonon bands almost vanishes from Γ to X. To exclude that the concentration difference may have an effect, we perform a numerical experiment: we start from the cubic structure doped at 0.11e/f.u. concentration (space group $Pm\bar{3}m$), and then we impose a slight (001) compressive bi-axial 0.8% strain by fixing the two in-plane lattice constants ($a$ and $b$) to a smaller value. This compressive strain makes the crystal structure of doped BaTiO₃ tetragonal and polar (space group $P4mm$). Figure 4a shows the optimized crystal structures of the two doped BaTiO₃. For doped BaTiO₃ at 0.11e/f.u. concentration, without strain, the ground state structure is cubic and the optimized lattice constant $a$ is 3.972 Å; under a 0.8% bi-axial (001) compressive strain, the ground state structure becomes tetragonal with the in-plane lattice constants $a$ and $b$ being fixed at 3.940 Å and the optimized long lattice constant $c$ being 4.019 Å. We find that the total electron-phonon coupling $\lambda$ increases from 0.50 in the $Pm\bar{3}m$ structure to 0.57 in the $P4mm$ structure. Figure 4b, c compare the imaginary part of the electron-phonon self-energy $\mathrm{Im}\Pi_{\mathbf{q}\nu}$ and the mode-resolved electron-phonon coupling $\lambda_{\mathbf{q}\nu}$ along the Γ → X path for the two doped BaTiO₃. Similar to Fig. 3a and b, we find that there is a notable difference in $\mathrm{Im}\Pi_{\mathbf{q}\nu}$ from the acoustic

phonon bands. The difference in $\mathrm{Im}\Pi_{\mathbf{q}\nu}$ is further "amplified" by the low phonon frequencies $\omega_{\mathbf{q}\nu}$, which results in

$$\text{without epitaxial strain } \lambda_{\text{acoustic}} = 0.27$$
$$\text{under 0.8\% (001) compressive strain } \lambda_{\text{acoustic}} = 4.45 \quad (6)$$

At the same time, we find that for polar modes,

$$\text{without epitaxial strain } \lambda_{\text{polar}} = 5.58$$
$$\text{under 0.8\% (001) compressive strain } \lambda_{\text{polar}} = 3.21 \quad (7)$$

This shows that under 0.8% (001) compressive strain, $\lambda_{\text{polar}}$ remains substantial (albeit reduced by about 40%), but $\lambda_{\text{acoustic}}$ is increased by one order of magnitude, which altogether leads to an enhancement of the total electron-phonon coupling $\lambda$. Note that in the numerical experiment, the two doped BaTiO₃ have exactly the same electron concentration, indicating that the additional increase in $\mathrm{Im}\Pi_{\mathbf{q}\nu}$ of the acoustic phonons arises solely from the crystal structure difference. A possible explanation, which is based on our calculations, is that in the cubic structure, some electron-phonon vertices $g_{ij}^\nu(\mathbf{k}, \mathbf{q})$ are exactly equal to zero because some atoms are frozen in the acoustic phonons, while in the low-symmetry structure, those $g_{ij}^\nu(\mathbf{k}, \mathbf{q})$ become non-zero. Because $\mathrm{Im}\Pi_{\mathbf{q}\nu} \propto |g_{ij}^\nu(\mathbf{k}, \mathbf{q})|^2$[48,49], this leads to an increase in $\mathrm{Im}\Pi_{\mathbf{q}\nu}$. In addition, the frequencies of acoustic phonon modes $\omega_{\mathbf{q}\nu}$ are very small and $\lambda_{\mathbf{q}\nu} \propto \mathrm{Im}\Pi_{\mathbf{q}\nu}/\omega_{\mathbf{q}\nu}^2$, therefore even a slight increase in $\mathrm{Im}\Pi_{\mathbf{q}\nu}$ results in a substantial enhancement in $\lambda_{\mathbf{q}\nu}$ (see Supplementary Note 16 for the demonstration of a specific acoustic phonon). Our numerical experiment also implies that in

doped $BaTiO_3$, when the electron concentration is close to the critical value, a small (001) compressive strain that lowers the crystal symmetry may also enhance its superconducting transition temperature due to the increased electron-phonon coupling, similar to doped $SrTiO_3$[18,22].

## Discussion

Finally we discuss possible experimental verification. Chemical doping[44,45,50–54] and epitaxial strain[55,56] have been applied to ferroelectric materials such as $BaTiO_3$. La-doped $BaTiO_3$ has been experimentally synthesized. High-temperature transport measurements show that $Ba_{1−x}La_xTiO_3$ exhibits polar metallic behavior but ultra-low-temperature transport measurements are yet to be performed[50–54]. We note that La doping in $BaTiO_3$ may result in some chemical disorder. While the randomness of La distribution in $La_xBa_{1−x}TiO_3$ may affect the transport properties in the normal state, Anderson's theorem asserts that superconductivity in a conventional superconductor is robust with respect to non-magnetic disorder in the host material[57]. As a consequence, the superconducting transition temperature $T_c$ of a conventional superconductor barely depends on the randomness of defects. In our case, the superconductivity in doped $BaTiO_3$ is phonon-mediated (i.e., conventional) and La is a non-magnetic dopant. Therefore Anderson's theorem applies and we expect that even if chemical disorder may arise in actual experiments, it does not affect the superconducting properties of doped $BaTiO_3$. In addition, we perform supercell calculations which include real La dopants. We find that even in the presence of real La atoms, the conduction electrons on Ti atoms are almost uniformly distributed in $La_xBa_{1−x}TiO_3$ (see Supplementary Note 8 and Supplementary Note 9 for details). Since our simulation does not consider dopants explicitly, a more desirable doping method is to use electrostatic carrier doping[58–60], which does not involve chemical dopants and has been successfully used to induce superconductivity in $KTaO_3$[61]. We clarify two points concerning the electrostatic doping method. (1) The electrostatic gating by ionic liquid can achieve a two-dimensional carrier density as high as $8 × 10^{14}$ cm$^{−2}$[62]. The induced electrons are usually confined in a narrow region that is a few nanometers from the surface/interface, which leads to an effective three-dimensional carrier density of about $1 × 10^{21} ∼ 5 × 10^{21}$ cm$^{−3}$[61,63]. In our current study, the critical concentration of doped $BaTiO_3$ is about $1.6 × 10^{21}$ cm$^{−3}$, which is feasible by this approach. (2) While the electrostatic doping method induces the carriers in the surface/interface area, we show that our results on bulk doped $BaTiO_3$ can still be used as a guidance to search for superconductivity in the surface area of $BaTiO_3$. We perform calculations of $Pt/BaTiO_3$ interface (see Supplementary Note 10) and find that just in the second unit cell of $BaTiO_3$ from the interface, the Ti-O displacement saturates and a bulk-like region emerges with almost uniform cation displacements. In addition, we calculate the electron-phonon properties of bulk $KTaO_3$ at 0.14$e$/f.u. doping (based on the experiment[61]) (see Supplementary Note 15). We find that the total electron-phonon coupling of $KTaO_3$ at 0.14$e$/f. u. doping is 0.36. Using McMillian equation (take $\mu^* = 0.1$) as a rough estimation of superconducting transition temperature $T_c$, we obtain a $T_c$ of about 68 mK, which is in reasonable agreement with the experimental value of 50 mK. While there is definitely room for improvement, our results demonstrate that for a given target material, its desirable bulk electron-phonon property can point to the right direction in which superconductivity is found in surface/interface regions.

In summary, we use first-principles calculations to demonstrate a large modulation of electron-phonon coupling and an emergent superconducting "dome" in $n$-doped $BaTiO_3$. Contrary to Anderson/Blount's weak electron coupling mechanism for "ferroelectric-like metals"[32–34], our calculations find that the soft polar phonons are strongly coupled to itinerant electrons across the polar-to-centrosymmetric phase transition in doped $BaTiO_3$ and as a consequence, the total electron-phonon coupling increases around the critical concentration. In addition, we find that lowering the crystal symmetry of doped $BaTiO_3$ by imposing epitaxial strain can also increase the electron-phonon coupling via a sizable coupling between acoustic phonons and itinerant electrons. Our work provides an experimentally viable method to modulating electron-phonon coupling and inducing phonon-mediated superconductivity in doped strong ferroelectrics. Our results indicate that the weak electron coupling mechanism for "ferroelectric-like metals"[32–34] is not necessarily present in doped strong ferroelectrics. We hope that our predictions will stimulate experiments on doped ferroelectrics and search for the phonon-mediated superconductivity that is predicted in our calculations.

## Methods

We perform first-principles calculations by using density functional theory[64–67] as implemented in the Quantum ESPRESSO package[68]. We use norm-conserving pseudo-potentials[69] with local density approximation as the exchange-correlation functional. For electronic structure calculations, we use an energy cutoff of 100 Ry. We optimize both cell parameters and internal coordinates in atomic relaxation, We find that the optimized crystal structures are in good agreement with experiments (see Supplementary Note 1). The detailed structural information is reported in Supplementary Note 7. In the strain calculations, the in-plane lattice constants are fixed while the out-of-plane lattice constant and internal coordinates are fully optimized. The electron Brillouin zone integration is performed with a Gaussian smearing of 0.005 Ry over a Γ-centered $\mathbf{k}$ mesh of $12 × 12 × 12$. The threshold of total energy convergence is $10^{−7}$ Ry; self-consistency convergence is $10^{−12}$ Ry; force convergence is $10^{−6}$ Ry/Bohr and pressure convergence for variable cell is 0.5 kbar. For phonon calculations, we use density functional perturbation theory[66] as implemented in the Quantum ESPRESSO package[68] (see Supplementary Note 11 for the validation of this method on a prototypical oxide $SrTiO_3$). The phonon Brillouin zone integration is performed over a $\mathbf{q}$ mesh of $6 × 6 × 6$. For the calculations of electron-phonon coupling and superconducting gap (see Supplementary Note 2), we use maximally localized Wannier functions and Migdal-Eliashberg theory, as implemented in the Wannier90[70] and EPW code[71]. The Fermi surface of electron-doped $BaTiO_3$ is composed of three Ti $t_{2g}$ orbitals. We use three maximally localized Wannier functions to reproduce the Fermi surface. The electron-phonon matrix elements $g^\nu_{ij}(\mathbf{k}, \mathbf{q})$ are first calculated on a coarse $12 × 12 × 12$ $\mathbf{k}$-grid in the electron Brillouin zone and a coarse $6 × 6 × 6$ $\mathbf{q}$-grid in the phonon Brillouin zone, and then are interpolated onto fine grids via maximally localized Wannier functions. The fine electron and phonon grids are both $50 × 50 × 50$. We check the convergence on the electron $\mathbf{k}$-mesh, phonon $\mathbf{q}$-mesh and Wannier interpolation and no significant difference is found by using a denser mesh. Details can be found in Supplementary Note 4. We solve a three-orbital Eliashberg equation to estimate the superconducting transition temperature $T_c$ (see Supplementary Note 3).

We only use Eliashberg equation when electron doping concentration is high enough so that $\lambda T_D/T_F < 0.1$ and Migdal's theorem is valid[29] ($\lambda$ is electron-phonon coupling, $T_D$ is Debye temperature and $T_F$ is Fermi temperature). Validation test of Migdal's theorem is shown in Supplementary Note 5.

We solve a three-orbital Eliashberg equation to estimate the superconducting transition temperature $T_c$. This method is compared to McMillan Equation. Details of Eliashberg Equation and McMillan Equation can be found in Supplementary Note 3.

**Reporting summary**. Further information on research design is available in the Nature Research Reporting Summary linked to this article.

## Data availability

The data that support the findings of this study are available from the corresponding author upon reasonable request.

## Code availability

The electronic structure calculations were performed using the open-source code Quantum Espresso[68]. Quantum Espresso package is freely distributed on academic use under the Massachusetts Institute of Technology (MIT) License.

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

## Acknowledgements

We acknowledge useful discussion with Kevin Garrity, Jia Chen and Jin Zhao. H.C. is supported by the National Natural Science Foundation of China under Project No. 11774236 and NYU University Research Challenge Fund. J.M. is supported by the Student Research Program in Physics of NYU Shanghai. NYU high performance computing at Shanghai, New York and Abu Dhabi campuses provide the computational resources.

## Author contributions

H.C. conceived and supervised the project. J.M. and H.C. performed the calculations. R.Y. contributed to the data analysis. H.C. and J.M. wrote the paper and all the authors commented on the paper.

## Competing interests

The authors declare no competing interests.
