## [Peer Review File · Nature Communications]

REVIEWER COMMENTS

Reviewer #1 (Remarks to the Author):

This theoretical paper presents first-principles calculations which predict a superconducting dome in doped BaTiO₃ near a structural transition induced by metallicity. The prediction is interesting and the work deserves to be published, however it inspires numerous critical comments.

- i. First of all, the capacity of first-principles theory to predict accurate T_c is far from established. As the case of MgB₂ has shown, calculations of superconducting critical temperature are impressive in postdiction (and not in prediction) because of the crucial role of subtle details in electron-phonon coupling.
- ii. In the present case, the authors begin by stating: “previous studies found that in n-doped BaTiO₃, increasing the carrier density gradually reduces its polar distortions and induces a continuous polar-to-centrosymmetric phase transition... [31, 32].” Ref. 31 and 32 are theoretical papers. Nevertheless, the authors write further: “It is evident that the critical concentration n_c of doped BaTiO₃ is 0.10e/f.u. (about $1.6 \cdot 10^{21} \text{ cm}^{-3}$).” The experimental situation is far less clear cut. According to ref. 46, a sample with a carrier density of $2 \cdot 10^{21} \text{ cm}^{-3}$ “remained paraelectric (cubic) down to the minimum accessible temperature of 2 K”. According to figure 1 of ref. 46 indicates the resistivity of this sample **increases** with cooling down to 2 K. Of course, this does not exclude the idea proposed in this paper (a superconducting transition below 2 K). However, I invite the authors to share this information with their readers.
- iii. It is easy to see that a comparison with Ca-doped strontium titanate has been the driving idea behind this study. However, even in pure strontium titanate, the accurate phonon spectrum is hard to capture by first principle calculations. For example, as seen in Aschauer & Spaldin Journal of Physics: Condensed Matter 26 , 122203 (2014), DFT calculations find imaginary frequencies. The experimental discoveries of both the ferroelectric order in Ca-doped strontium titanate (Bednorz & Müller, Phys. Rev. Lett. 52, 2289 (1984) and polar-to-centrosymmetric transition (ref. 20) were not anticipated by theory.
- iv. Superconductivity in doped strontium titanate is restricted to a finite doping window of $1e-5 < n < 0.02$. One mystery is the total vanishing of superconductivity above 0.02 carrier f. u. in spite of increasing density of states. Looking at fig. 1f, one sees that this theory predicts that in doped BaBiO₃, T_c remains finite below 0.06 and above 0.14. Why then it vanishes so abruptly in moderately dense doped SrTiO₃?

These remarks do not disqualify this paper as an interesting attempt to predict superconductivity based on an Eliashberg theory of coupling of electrons with a soft phonon mode. But, I think that the main conclusion is an interesting speculation waiting experimental verification.

Reviewer #2 (Remarks to the Author):

This manuscript reported a calculation on superconductivity induced by strong electron-soft phonon coupling in doped strong ferroelectrics. As far as I know, the e-soft ph coupling and the superconductivity across the critical point of polar-nonpolar transition was barely calculated. From this point of view, this work is intriguing. The authors used BaTiO₃ as a template material. However, it is still under debate whether doped BaTiO₃ is a polar metal or not. This manuscript can't meet the criterion of Nature Communications at least in the present form. The follow questions should be addressed:

1) Neutron scattering measurement showed that the polar insulating phase and non-polar conducting phase are spatially separated in BTO (Jeong, PRB 84, 064125 (2011)). Do the authors have any comments on it?

2) The authors predicted that BTO may show superconductivity at low temperatures. However, it was never observed experimentally. The authors attributed it to the poor low temperature measurements in previous reports. In my opinion, this statement is oversimplified. We know that for La doped BTO, different papers reported distinct transport behaviors for the same La content (also see ref.44, 45 in the manuscript). I guess the randomness of La distribution may affect the transport behaviors in electron doped BTO. Can the authors give an additional discussion about how the possible superconducting state will be influenced by the random doping effect? It would be more instructive to experimentalists.

3) The authors also suggested the electrostatic effect as a possible route to achieve uniform electron doping. However, the maximum doping from ion liquid gating is only about $10^{20} - 10^{21} \text{ cm}^{-3}$, that is too low to meet the requirement of doping levels in this manuscript.

4) Related to the previous question, the electrostatic doping is a method to introduce surface carriers. It cannot induce bulk electrons doping. My question is that if the superconductivity is induced by doping on the surface of BTO, can it be related to the bulk phonon properties and why?

5) In ref. 20 of the manuscript, the superconducting critical temperature is increased near the 'FE' critical point for Ca:SrTiO₃. However, the 'FE' critical doping level is not located at the optimal doping level of the superconducting dome, which is different from the calculation of the authors. Do the authors have any comments?

Reviewer #3 (Remarks to the Author):

Review of NCOMMS-20-35777, "A large modulation of electron-phonon coupling and an emergent superconducting dome in doped strong ferroelectrics" by Jiaji Ma et al.

This theoretical study investigates electron-phonon coupling and the emergence of phonon-driven superconductivity in doped strong ferroelectrics. The authors choose to focus on the canonical strong ferroelectric BaTiO₃ (perovskite compound) and present the role of electron doping on phonon softening and the polar instability in cubic and tetragonal phases. While the study will be of relevance to the much-studied superconducting transition in doped SrTiO₃, it is also interesting because this case corresponds to a much higher critical carrier density. BaTiO₃ is a wide bandgap ferroelectric

material and is well known to exhibit unusually large Born effective charges of oxygen, indicating large polarizability and dielectric constant. The polar instability in this compound is attributed to strong p-d hybridization (also known as second-order Jahn Teller distortion). The results from this study are intriguing (e.g. reporting a substantial el-ph coupling constant of 0.6, as well as an opportunity to increase T_c with applied strain) and should motivate further investigations of the coupling between soft polar modes and itinerant electrons in doped oxides near polar phase instabilities.

While the manuscript is interesting and appears sound overall, I have several comments and requests that need to be addressed before the manuscript could be considered for publication.

1) In Fig 1, I am wondering why the orthorhombic phase does not appear to become stabilized when increasing doping concentration. Instead the system directly changes from rhombohedral to tetragonal and cubic. Could the authors comment on that?

2) I would like the authors to clarify how the electron doping impacts the p-d hybridization and Born effective charges. It seems that e- doping might screen the effective interaction between Ti-O and tune the instability. Also, it would be great if the author could also discuss the phonon behavior upon doping, especially in the rhombohedral phase.

3) I am also wondering if the author observed any localization behavior of the doped electron. In this case, added electrons (I am not sure if they are really itinerant) follows the d- band character, maybe they would also contribute to magnetism.

4) The authors did not mention how the structure (lattice parameters and ionic coordinates) are changing with electron doping. Could they please provide that information?

5) The authors should try to provide a possible explanation for the large-electron phonon coupling involving the acoustic branches in the tetragonal phase, contrasting with the behavior in the cubic phase.

6) How did the authors conclude that the phonons are mainly coupled with itinerant electrons? I am wondering if the author could compare the electronic DOS with different amplitudes of polar phonon distortions to explicitly show the coupling between itinerant electrons and the polar phonon mode.

7) In the free energy diagram at $T=0$ K, does the Y-axis represent the internal energy or enthalpy? In the case of complete relaxation, internal energy and enthalpy are nearly equal, while if author performed the fixed volume/structure calculation then the PV term is important.

8) The discussion of potential comparison with experiments would deserve to be expanded. How robust do the authors believe their conclusions are if one is to include explicit chemical disorder (dopants or vacancies)? The authors mention that comparison with electrostatic doping experiments could be easier or more direct, and point out (Ref 53) the reported observation of superconductivity in KTaO_3 upon electrostatic carrier doping. If so, why not perform/extend the theory to the case of KTaO_3 , which could already afford a direct test?

Reviewer #4 (Remarks to the Author):

I find the paper "A large modulation of electron-phonon coupling and an emergent superconducting dome in doped strong ferroelectrics" to be timely and potentially relevant for generation of interest in the ferroelectrics as a playground for finding possible new superconductors. I am recommending

substantial revision before a decision on acceptance or no-acceptance can be made.

Here are my comments to the authors

1. "previous studies found that in n-doped BaTiO₃, increasing the carrier density gradually reduces its polar distortions and induces a continuous polar-to-centrosymmetric phase transition (similar to Sr_{1-x}Ca_xTiO_{3-δ})"

Is the picture for the phase transition in Ca-STO that simple? I think it is a bit more involved than that.

2. "the critical concentration for the phase transition is about 10²¹/cm³, which is high enough so that the electron-phonon coupling can be directly calculated within the Migdal's approximation"

I don't think the comparison between the Ca case and other cases is that straightforward. Also, this seems like cherry picking in terms of very large doping range in STO for which it superconducts.

3. "Motivated by the above experiments and theories "

The authors should cite other experimental works on STO, some are listed here M. N. Gastiasoro et al., Superconductivity in Dilute SrTiO₃: A Review, Ann. Phys. 168107 (2020)

4. "superconductivity emerges at a much lower carrier concentration 10¹⁷-10²⁰ /cm³, which invalidates the Migdal's approximation and Eliashberg equation "

The argument should be spelled out more precisely in terms of the energy scales that are being compared with each other. And what are the actual experimental magnitudes for those compared values? Some accuracy won't hurt here.

"we find that the phonon bands associated with the soft polar optical phonons are strongly coupled to itinerant electrons across the polar-to-centrosymmetric phase transition in doped BaTiO₃ "

5. What's the insight for "why" this is the case?

"In addition, we find that close to the critical concentration, lowering the crystal symmetry of doped BaTiO₃ by imposing epitaxial strain further increases the superconducting temperature via a sizable coupling between itinerant electrons and acoustic phonon bands."

What about the ferroelectric phonon to electron coupling change with strain?

6. "Our results show that the weakly coupled electron mechanism in "ferroelectric-like metals" is not necessarily present in doped strong ferroelectrics and as a consequence, the soft polar phonons can be utilized to induce phonon-mediated superconductivity across a structural phase transition."

This needs more explanation for why "is not necessary present". Right now, it's just a statement but not an explanation, which would be helpful to have here or to come back to this point later in the text to explain "why".

7. "Fig. 1a shows that as electron doping concentration n increases from 0 to 0.15e/f.u., BaTiO₃ transitions from the rhombohedral structure to the tetragonal structure, and finally to the cubic structure."

Is the role of disorder due to dopants considered or derived here? How is its relevance in real

materials ruled out? Any experimental study that could be cited?

8. "Furthermore the low electron concentration in the rhombohedral structure invalidates Migdal's theorem and electron-phonon coupling can not be calculated within Migdal's approximation. "

The authors should probably spell out which values compared to which to support this statement, not clear. (It is in the SM but some readers may not get there, or provide a pointer).

9. "that inversion symmetry breaking by collective polar displacements in a metal relies on a weak coupling between itinerant electrons and soft phonons responsible for removing inversion symmetry. "

I know that this is being cited in the literature as "weak" vs. "strong" coupling regularly, but it would be nice to quantify the statements like this, what is weak and what is strong limit? Can we define for the readers' benefit and also for the accuracy of the statements?

10. "the phonon bands associated with the zone-center polar phonons have the strongest coupling to itinerant electrons, while the couplings of other phonon bands are much weaker. "

This sounds like a relative arguments, can it be made more precise here in terms of the actual numbers?

11. "and therefore can also make non-negligible contribution to the total electron-phonon coupling " Provide a number, it's hard to judge the acoustical phonons' contribution based on the plots in Fig 3. I cannot really see the amplification effect. Can it be also made more quantitative statement?

12. "a thought-experiment "

Should probably be "a numerical experiment" in several places.

13. " λ increases from 0.50 in the P m3m structure to 0.57 in the P 4mm structure, and the superconducting transition temperature T_c increases from 0.76 K in the P m3m structure to 2.0 K in the P 4mm structure "

I don't know if we can trust "Eliashberg equation and McMillan's formula" with such small change in λ and draw conclusions that 2 K is 2 K, it might be 2 might be 10 in real material. I'd like to see some comment about the accuracy of the expectation. These predictions are quite often off by a numerical factor.

14. "The difference in $\text{Im}\Pi_{\text{qv}}$ is further "amplified" by the low phonon frequencies ω_{qv} . Fig. 4c explicitly compares the mode-resolved electron-phonon coupling λ_{q1} for the lowest phonon band of the two doped BaTiO₃. It is evident that λ_{q1} is substantially larger in $\sim 0.11\text{e/f.u.}$ concentration (space group Pm3m), and then we impose a slight compressive the optimized lattice constant a is 3.972 Å; under a 0.8% biaxial compressive strain, the ground state structure becomes tetragonal with the optimized short lattice constant a being electron-phonon coupling λ increases from 0.50 in the P m3m structure to 0.57 in the P 4mm structure, and the superconducting transition temperature T_c increases from 0.76 K in the \sim the P 4mm tetragonal structure than in the P m3m cubic structure. "

I'd like to see the mode-resolved coupling for the FE modes next to the acoustic ones. How is the FE one changed by strain?

What is the scale of this mode-resolved λ ? Is it not the same as that of the averaged one?

15. "the exactly same electron concentration, indicating that the additional increase in $\text{Im}\Pi_{\text{qv}}$ arises solely from the crystal structure difference. Lowering the symmetry of BaTiO₃ crystal structure allows more electron-phonon scattering processes that would be forbidden in the cubic structure by symmetry considerations. "

Is this a quantitative or qualitative statement? I suggest quoting numbers. Is this general or specific to these calculations? Seems to be too important to pass through so quickly - need more clarification and quantitative argumentation beyond "the plots show". Is there a more general insight to be learned?

16. "a small compressive strain that lowers the crystal symmetry can also increase its superconducting transition temperature, similar to doped SrTiO₃"

Please specify compressive strain in which direction. Also please see other relevant experiments in ref. M. N. Gastiasoro, Superconductivity in Dilute SrTiO₃: A Review, Ann. Phys. 168107 (2020). In terms of the polar modes and anisotropies the strain direction is important.

17. "Since our simulation does not consider dopants explicitly, a more desirable doping method is to use electrostatic carrier doping "

But then the structure doesn't not change. So how does electrostatic gating of the surface should work in terms of the structural phase transition? Are the authors saying that just strain and surface conductivity enough? Please clarify with more precision.

18. "We hope that our predictions will stimulate new experiments on doped ferroelectrics and, if confirmed, may help shed light on the mysterious origin of the superconductivity in doped SrTiO₃."

I don't think the projection for STO is that simple, given that the authors say that for their material Migdal's approximation is valid, while for STO it is not that straightforward. I think the overall projected connection to STO is somewhat premature, many more details would need to be covered to make any such conclusion.

Reply to Reviewers (NCOMMS-20-35777)

We would like to thank all the four reviewers for their helpful questions and comments. In the process of answering these questions/comments and revising the text, our manuscript has improved. Below, we address all their concerns in detail. The resulting modifications to the manuscript clarify some of the most important points of our work.

For each reviewer, we address each question/comment by first quoting the question/comment followed by our reply.

Response to Reviewer #1

We thank the reviewer for her/his comments that “the prediction is interesting and the work deserves to be published.”

1) First of all, the capacity of first-principles theory to predict accurate T_c is far from established. As the case of MgB_2 has shown, calculations of superconducting critical temperature are impressive in postdiction (and not in prediction) because of the crucial role of subtle details in electron-phonon coupling.

We agree with the reviewer that accurately predicting superconducting transition temperature T_c has always been a daunting task. Even for conventional phonon-mediated superconductors, there is uncertainty in the Migdal-Eliashberg theory such as Morel-Anderson pseudopotential μ^* . As we show in the inset of Fig. 3f in the main text, T_c strongly depends on the value of μ^* .

However, the calculation of electron-phonon coupling λ is more reliable, as long as Migdal approximation is valid [1]. In our calculations, the main conclusion is that the electron-phonon coupling λ of doped $BaTiO_3$ can be modulated by carrier density n due to the strong coupling between soft polar phonons and itinerant electrons. Around the polar-to-centrosymmetric structural phase transition, the polar phonon gets softened and λ is increased. As a consequence of this, a phonon-mediated superconducting dome emerges

around the critical concentration n_c . Our first-principles calculations support this picture. As for the accurate value of superconducting transition temperature T_c , we agree with the reviewer that there is uncertainty in our estimation due to μ^* and other subtle technical details. That is exactly why we plot T_c as a function of μ^* to get a range of T_c in our estimation.

On the other hand, we would like to mention that recently a couple of *ab initio* predictions of phonon-mediated superconductors have been later confirmed in experiments (see Table R1). While there is still much room for improving the accuracy of the predicted T_c , it shows that first-principles calculations of electron-phonon coupling can point to the right direction of searching for new phonon-mediated superconductors.

In the revised version, we add text on Page 9 (highlighted in blue text) to stress that the predicted superconducting temperature T_c has uncertainty due to μ^* and other technical details, but the picture of an increased electron-phonon coupling around the structural phase transition in doped BaTiO₃ is robust.

TABLE R1: Comparison of the superconducting transition temperatures T_c between first-principles predictions and experiments.

	first-principles predictions (K)	experimental values
SiH ₄	39 [2]	17.5 [3]
H ₃ S	191-204 [4]	203 [5]
LaH ₁₀	238 [6]	250 [7]

2) In the present case, the authors begin by stating: “previous studies found that in n-doped BaTiO₃, increasing the carrier density gradually reduces its polar distortions and induces a continuous polar-to-centrosymmetric phase transition... [31, 32].” Ref. 31 and 32 are theoretical papers. Nevertheless, the authors write further: It is evident that the critical concentration n_c of doped BaTiO₃ is 0.10e/f.u. (about $1.6 \times 10^{21} \text{cm}^{-3}$). The experimental situation is far less clear cut. According to ref. 46, a sample with a carrier density of $2 \times 10^{21} \text{cm}^{-3}$ remained paraelectric (cubic) down to the minimum accessible temperature of 2 K. According to figure 1 of ref. 46 indicates the resistivity

of this sample increases with cooling down to 2 K. Of course, this does not exclude the idea proposed in this paper (a superconducting transition below 2 K). However, I invite the authors to share this information with their readers.

We apologize for the confusion on this issue and we thank the reviewer for pointing this out to us.

We clarify here that we determine the critical concentration n_c of doped BaTiO₃ using our own theoretical calculations, i.e. the concentration at which the polar displacements of BaTiO₃ just vanish. Our result ($n_c = 0.1e/f.u. \simeq 1.6 \times 10^{21} \text{cm}^{-3}$) is consistent with the previous *theoretical* studies [8, 9] and is also close to the experiment (PRL 104, 147602 (2010)) in which the critical concentration is found to be about $1.9 \times 10^{21} \text{cm}^{-3}$.

The reviewer mentions that in PRL 104, 147602 (2010) (Ref. 46 in the previous version of main text), a sample with a carrier density of $2.0 \times 10^{21} \text{cm}^{-3}$ remains paraelectric (cubic) down to the minimum accessible temperature of 2 K and the resistivity of this sample increases with cooling down to 2K. We explain that the concentration in this sample is larger than the critical value $n_c = 1.9 \times 10^{21} \text{cm}^{-3}$, so we do expect that this sample has a cubic structure. Our first-principles calculations also find that doped BaTiO₃ at the same carrier density ($n = 2.0 \times 10^{21} \text{cm}^{-3}$) has a cubic ground-state structure. We agree with the reviewer that the resistivity of that sample increases upon cooling, which deviates from a standard metallic behavior. The authors of PRL 104, 147602 (2010) did not explain the origin of this transport anomaly. However, we notice that the resistivity increases slowly upon cooling, implying that an energy gap is not opened (otherwise the resistivity would exhibit an exponential increase). The slow increase of resistivity might be due to weak localization, but high quality samples are needed to further clarify this important point.

In the revised version, we change the text from “consistent with previous studies” to “consistent with the previous theoretical studies” and add text on Page 4 and 5 (highlighted in blue text) to include information about the experimental results of doped BaTiO₃.

3) It is easy to see that a comparison with Ca-doped strontium titanate has been the driving idea behind this study. However, even in pure strontium

FIG. R1: **a)** Phonon band structure of cubic SrTiO₃ (space group No. 221 $Pm\bar{3}m$). A \mathbf{k} -point path including the high-symmetry points $\Gamma(0,0,0)$ -X(0.5,0,0)-M(0.5,0.5,0)-R(0.5,0.5,0.5)- $G(0,0,0)$ is used in the calculation. **b)** Phonon band structure of tetragonal SrTiO₃ with an out-of-phase oxygen octahedral rotation (space group No. 140 $I4/mcm$). A \mathbf{k} -point path including the high-symmetry points $\Gamma(0,0,0)$ -X(0,0,0.5)-Y(-0.246,0.246,0.5)- $\Sigma(-0.373,0.373,0.373)$ - $\Gamma(0,0,0)$ -Z(0.5,0.5,-0.5)-Y'(0.5,0.5,-0.246)- $\Sigma'(0.373,0.627,-0.373)$ -N(0,0.5,0)-P(0.25,0.25,0.25)- $\Gamma(0,0,0)$ is used in the calculation.

titanate, the accurate phonon spectrum is hard to capture by first principle calculations. For example, as seen in Aschauer and Spaldin *Journal of Physics: Condensed Matter* **26**, 122203 (2014), DFT calculations find imaginary frequencies. The experimental discoveries of both the ferroelectric order in Ca-doped strontium titanate (Bednorz and Miller, *Phys. Rev. Lett.* **52**, 2289 (1984) and polar-to-centrosymmetric transition (ref. 20) were not anticipated by theory.

We thank the reviewer for this important comment.

We first explain that pure SrTiO₃ is cubic at room temperature and develops an antiferrodistortive (AFD) rotation below 105 K (an out-of-phase oxygen octahedral rotation about the c -axis, $a^0a^0c^-$ in the Glazer notation) [10, 11]. In *Journal of Physics: Condensed Matter* **26**, 122203 (2014), Aschauer and Spaldin use a 5-atom cell and cubic symmetry is imposed in their calculations, therefore the AFD rotation is artificially forbidden. This is exactly why we see imaginary phonon frequencies at the R point. If a larger cell is used which can accommodate the AFD rotation, one will see the imaginary

phonon frequencies disappear. To demonstrate this point, we perform additional calculations. Fig. R1a shows the phonon band structure of cubic SrTiO₃ (space group $Pm\bar{3}m$, No. 221). Consistent with Aschauer and Spaldin’s calculations, imaginary phonon modes appear at M and R points. The imaginary phonon mode at R point corresponds to the AFD rotation ($a^0a^0c^-$ Glazer tilts). The unstable phonon mode at M point corresponds to an energetically less favorable $a^0a^0c^+$ Glazer tilts. Fig. R1b shows the phonon band structure of the SrTiO₃ crystal structure with the AFD rotation (space group $I4/mcm$, No. 140), where the imaginary phonon modes disappear.

Second, for Ca_{*x*}Sr_{1-*x*}TiO₃, ferroelectric order occurs when Ca concentration is in the dilute limit ($0.002 < x < 0.02$). This requires a huge supercell for direct first-principles modelling, which goes well beyond our computational capability. However, molecular dynamics may be able to directly simulate Ca_{*x*}Sr_{1-*x*}TiO₃ ($0.002 < x < 0.02$), which warrants further investigation in future studies.

Third, we agree with the reviewer that the work on Ca-doped SrTiO₃ [12] is one of the beautiful experiments that inspires us to study doped BaTiO₃. However, the origin of the superconductivity that emerges in doped SrTiO₃ is still controversial, which motivates us to study possible phonon-mediated superconductivity in doped strong ferroelectrics (BaTiO₃ as a prototype). The physical properties of strong ferroelectrics, such as polarization, Born effective charges and phonons are well described by first-principles calculations [13–15].

In the revised version, we add a new section in the Supplementary Materials (Section XI “Phonon dispersion of undoped SrTiO₃”) to show that first-principles calculations are capable to get the correct phonon spectrum of pure SrTiO₃. We also cite the paper Phys. Rev. Lett. **52**, 2289 (1984) (now Reference 26 in the main text) that the reviewer brought to our attention.

4) Superconductivity in doped strontium titanate is restricted to a finite doping window of $1e - 5 < n < 0.02$. One mystery is the total vanishing of superconductivity above 0.02 carrier f.u. in spite of increasing density of states. Looking at fig. 1f, one sees that this theory predicts that in doped BaTiO₃, Tc remains finite below 0.06 and above 0.14. Why then it vanishes so abruptly

FIG. R2: Superconducting critical temperature T_c as a function of carrier density n for $\text{Sr}_{1-x}\text{La}_x\text{Ti}({}^{16}\text{O}_{1-z}{}^{18}\text{O}_z)_3$, $\text{SrTiO}_{3-\delta}$, and $\text{SrTi}_{1-x}\text{Nb}_x\text{O}_3$. This figure is adapted from Fig. 3a in Ref. [26].

in moderately dense doped SrTiO_3 ?

We thank the reviewer for this comment.

To our best knowledge, the reason why superconductivity in doped SrTiO_3 vanishes above a critical carrier density (sometimes called upper critical carrier density) is still not clear (we acknowledge private communications with Prof. Xiao Lin, the first author of Ref. [16] and [17]). There are a few theories [18–25] but consensus has not been reached. Experimentally, Fig. R2 (adapted from Fig. 3a of Ref. [26] for ease of reading) shows that the upper critical carrier density for superconductivity is different in oxygen-deficient $\text{SrTiO}_{3-\delta}$, La-doped SrTiO_3 and Nb-doped SrTiO_3 . This upper critical carrier density also depends on the concentration of ${}^{18}\text{O}$. Combining different experiments, it seems that the superconducting transition temperature T_c change gradually with the carrier density, in particular in oxygen-deficient $\text{SrTiO}_{3-\delta}$ [26]. We would also like to mention that Fig. 3f in our main text uses a linear scale for carrier density, while Fig. 6a of Ref. [17], Fig. 3a

of Ref. [26] and Fig. 4c of Ref. [27] all use a logarithmic scale for carrier density. This makes the T_c dependence on carrier density seem gradual in our figure, but more abrupt in Ref. [17, 26, 27].

In the revised version, we add text on Page 2 to mention that it is still not clear why superconductivity in doped SrTiO₃ vanishes above the upper critical carrier density, in spite of the increasing density of states [26].

Response to Reviewer #2

We would like to thank the reviewer for commenting that “as far as I know, the e-soft ph coupling and the superconductivity across the critical point of polar-nonpolar transition was barely calculated. From this point of view, this work is intriguing.”

1) Neutron scattering measurement showed that the polar insulating phase and non-polar conducting phase are spatially separated in BTO (Jeong, PRB 84, 064125 (2011)). Do the authors have any comments on it?

We thank the reviewer for bringing PRB 84, 064125 (2011) to our attention. We cite it in our revised version.

We agree with the authors of PRB 84, 064125 (2011) that in their samples of BaTiO_{3- δ} , the ferroelectric ordering and metallic conduction are two spatially separated phases. In fact, PRL 104, 147602 (2010), which we have cited in our main text, points to another issue concerning the structure of BaTiO_{3- δ} : when the concentration of oxygen vacancies exceeds a critical value in BaTiO_{3- δ} , the perovskite structure can no longer be stabilized and transforms into a hexagonal polymorph (space group $P6_3/mmc$).

In our study, we focus on the soft polar phonon of BaTiO₃, which mainly involves the movement of Ti and O atoms. While oxygen vacancies can induce free electrons into BaTiO₃, they inevitably break the TiO₆ oxygen octahedron and may affect the polar phonon. Therefore, we suggest that La-doping and ionic liquid gating are probably better methods of injecting electrons into BaTiO₃ because 1) in La _{x} Ba_{1- x} TiO₃ samples, phase

separation has not been reported in experiments [28, 29]; 2) the carriers induced by ionic liquid gating are confined in the surface layers and thus chemical disorder is minimized [30].

In the revised version, we cite the paper PRB 84, 064125 (2011) (now Reference 46 in the main text) and add text on Page 4 and 5 to provide more discussion about oxygen-deficient BaTiO₃.

2) The authors predicted that BTO may show superconductivity at low temperatures. However, it was never observed experimentally. The authors attributed it to the poor low temperature measurements in previous reports. In my opinion, this statement is oversimplified. We know that for La doped BTO, different papers reported distinct transport behaviors for the same La content (also see ref.44, 45 in the manuscript). I guess the randomness of La distribution may affect the transport behaviors in electron doped BTO. Can the authors give an additional discussion about how the possible superconducting state will be influenced by the random doping effect? It would be more instructive to experimentalists.

We thank the reviewer for this important comment.

We agree with the reviewer that in La-doped BaTiO₃, the randomness of La distribution may affect the transport properties in the normal state (the metallic state above the superconducting transition temperature). However, Anderson's theorem asserts that superconductivity in a conventional superconductor is robust with respect to non-magnetic disorder in the host material [31]. As a consequence, the superconducting transition temperature T_c of a conventional superconductor barely depends on the randomness of defects. In our case, the superconductivity in doped BaTiO₃ is phonon-mediated (i.e. conventional) and La is a non-magnetic dopant. Therefore Anderson's theorem applies and we expect that the superconducting phase in doped BaTiO₃ is robust against chemical disorder (though the normal state resistivity of doped BaTiO₃ depends on sample qualities [28, 29]).

On the other hand, we perform additional supercell calculations of La-doped BaTiO₃. We study four representative cases: a $2 \times 2 \times 2$ supercell with one Ba atom replaced by La to

FIG. R3: **a)** A $2 \times 2 \times 2$ supercell of La-doped BaTiO₃ (with one La atom in the supercell). **b)** The number of conduction electrons on each Ti atom in the $2 \times 2 \times 2$ supercell. **c)** A $3 \times 3 \times 3$ supercell of La-doped BaTiO₃ (with one La atom in the supercell). **d)** The number of conduction electrons on each Ti atom in the supercell. **e)** and **g)** A $3 \times 3 \times 3$ supercell of La-doped BaTiO₃ (with two La atoms in the supercell). **f)** and **h)** The number of conduction electrons on each Ti atom in the $3 \times 3 \times 3$ supercell.

simulate a carrier density of $0.125e/\text{f.u.}$, a $3 \times 3 \times 3$ supercell with one Ba atom replaced by La to simulate a carrier density of $0.037e/\text{f.u.}$ and two $3 \times 3 \times 3$ supercells with two Ba atoms replaced by La to simulate a carrier density of $0.074e/\text{f.u.}$ (the distribution of La atoms is different in the two $3 \times 3 \times 3$ supercells). Panel **a** of Fig. R3 shows the crystal

structure of supercells and panel **b** of Fig. R3 shows the number of conduction electrons on each Ti atom. In the three cases of $3 \times 3 \times 3$ supercell, the number of conduction electrons on Ti atoms has a tiny variation of $0.003e/\text{Ti}$; and in the case of $2 \times 2 \times 2$ supercell the variation is negligible ($< 10^{-7}e/\text{Ti}$). These results reveal that even in the presence of real La atoms, the conduction electrons on Ti atoms are almost uniformly distributed.

In the revised version, we modify our statement of low-temperature transport measurements on Page 10 and add text on Page 11 (highlighted in blue text) to carefully discuss the dopant randomness effect on transport properties of doped BaTiO_3 . We also add a section about the supercell calculation of La-doped BaTiO_3 (Section VIII) in the Supplementary Materials.

3) The authors also suggested the electrostatic effect as a possible route to achieve uniform electron doping. However, the maximum doping from ion liquid gating is only about $10^{20} - 10^{21} \text{cm}^{-3}$, that is too low to meet the requirement of doping levels in this manuscript.

We thank the reviewer for this comment.

Our calculations find that the optimal doping of BaTiO_3 is $0.1e/\text{f.u.}$, equivalent to a three-dimensional (3D) carrier density of $n_{3\text{D}} \simeq 1.6 \times 10^{21} \text{cm}^{-3}$. We argue that inducing this doping level by ionic liquid gating is feasible in experiment.

First, the authors of Ref. [30] use ionic liquid gating and achieve a 2D carrier density of $3.7 \times 10^{14} \text{cm}^{-2}$ (at $V_G = 6 \text{V}$) in KTaO_3 , which induces superconductivity in KTaO_3 at 50 mK. The authors find that almost all the carriers in KTaO_3 are confined within 2 nm from the surface (see Fig. R4, which is adapted from Fig. S5 of Ref. [30] for ease of reading), hence the 2D carrier density is translated into a 3D carrier density of $2.2 \times 10^{21} \text{cm}^{-3}$ or equivalently $0.14e/\text{f.u.}$

Second, at the well-known $\text{LaAlO}_3/\text{SrTiO}_3$ interface, PRL 102, 216804 (2009) shows that for a 2D carrier density of $3.3 \times 10^{14} \text{cm}^{-2}$, itinerant electrons are narrowly confined to the interface within a few nanometers, which leads to a local 3D carrier density about

FIG. R4: Depth d dependence of three-dimensional charge carrier density n_{3D} for two-dimensional carrier density n_{2D} . Right axis shows corresponding number of carriers per unit cell of KTaO_3 . Dashed lines correspond to a mean value of n_{3D} for each V_G . Shaded area corresponds to n_{3D} below $1.4 \times 10^{20} \text{ cm}^{-3}$ that can be achieved by chemical doping. Carriers are strongly confined towards the surface ($d=0$) with increasing gate bias. Inset shows confinement potential for various n_{2D} . This figure is adapted from Fig. S5 of Ref. [30].

$5 \times 10^{21} \text{ cm}^{-3}$ (equivalently about $0.3e/\text{f.u.}$) close to the interface.

Since the permittivity of BaTiO_3 is between that of KTaO_3 and SrTiO_3 , we expect that the itinerant electrons in doped BaTiO_3 will also be confined to the interface within a few nanometers (as mentioned by the reviewer). Ref. [32] shows that electrostatic gating by ionic liquid can achieve a two-dimensional carrier density as high as $8 \times 10^{14} \text{ cm}^{-2}$. This means that a 3D carrier density of about $1.6 \times 10^{21} \text{ cm}^{-3}$ can be achieved by ionic liquid gating in doped BaTiO_3 close to the interface.

In the revised version, we expand the Section Discussion (on Page 11 and 12, highlighted in blue text) in which we provide more details of electrostatic doping and show that our critical electron concentration of $1.6 \times 10^{21} \text{ cm}^{-3}$ is feasible by this approach.

4) Related to the previous question, the electrostatic doping is a method to introduce surface carriers. It cannot induce bulk electrons doping. My

FIG. R5: a) The crystal structure of Pt/BaTiO₃ heterostructure. The two Pt/TiO₂ interfaces are not symmetric due to the polarization in BaTiO₃ (the polarization points from left to right). The green, blue, red and gray balls are Ba, Ti, O and Pt atoms. b) The layer-resolved Ti-O displacement along the z -axis δ . The purple shade highlights the bulk-like region. c) The electron-phonon spectral function $\alpha^2 F(\omega)$ of BaTiO₃ at 0.09e/f.u. doping (top) and of KTaO₃ at 0.14e/f.u. doping (bottom). The dashed lines are the accumulative electron-phonon coupling.

question is that if the superconductivity is induced by doping on the surface of BTO, can it be related to the bulk phonon properties and why?

We thank the referee for this important comment.

We agree with the reviewer that the chemical environment of surface is generally different from bulk. However, we would like to provide two results which show that our calculations of bulk BaTiO₃ can also be used as a guidance to search for superconductivity in the surface area of BaTiO₃.

First we perform additional calculations to simulate a Pt/BaTiO₃ heterostructure grown

on a SrTiO₃ substrate. Pt is a representative electrode. We include 16 unit cells of BaTiO₃ and two Pt/TiO₂ interfaces. Fig. R5a shows the optimized crystal structure. The two interfaces are not equivalent due to the presence of BaTiO₃ polarization. Fig. R5b shows the layer-resolved Ti-O displacement δ in BaTiO₃ thin films. We find that at both interfaces, two or three atomic layers away from either interface a bulk-like region emerges in which δ is almost uniform. Electrostatic doping can induce itinerant electrons into a region of the target material that is a few nanometers from the surface/interface [30]. Our calculations indicate that other than the very few atomic layers in the vicinity of surface/interface, BaTiO₃ thin films exhibit bulk-like structural properties with uniform cation displacements.

Second Ref. [30] used electrostatic doping to induce superconductivity in the surface/interface region of KTaO₃. We perform additional calculations of *bulk* KTaO₃ and compare its electron-phonon spectral function $\alpha^2F(\omega)$ to BaTiO₃. Based on the experiment [30], we choose 0.14e/f.u. doping for KTaO₃. For BaTiO₃, we choose 0.09e/f.u. doping in the comparison. The result is shown in Fig. R5c. We find that the total electron-phonon coupling of KTaO₃ at 0.14e/f.u. doping is 0.36, while that of BaTiO₃ at 0.09e/f.u. doping is 0.61. To get a rough estimation of superconducting transition temperature T_c , we use McMillian equation (take $\mu^* = 0.1$) and obtain that T_c for doped KTaO₃ is 68 mK, which is in reasonable agreement with the experimental value of 50 mK. While there is definitely room for improvement, our additional calculations show that for a given target material, its desirable bulk electron-phonon property can provide useful guidance to search for superconductivity in surface/interface regions.

In the revised version, we expand the Section Discussion (on Page 11 and 12) in which we provide more details of electrostatic doping and emergent superconductivity in KTaO₃ induced by electrostatic doping. We also add two new sections in the Supplementary Materials: Section X discusses the Pt/BaTiO₃ heterostructures and shows that BaTiO₃ thin films can exhibit bulk-like structural properties except for the few atomic layers in the vicinity of interface/surface. Section XV shows the comparison between doped BaTiO₃ and doped KTaO₃.

5) In ref. 20 of the manuscript, the superconducting critical temperature is

increased near the FE critical point for Ca:SrTiO₃. However, the FE critical doping level is not located at the optimal doping level of the superconducting dome, which is different from the calculation of the authors. Do the authors have any comments?

We thank the reviewer for this intriguing comment.

We agree with the reviewer that in Ca-doped oxygen-deficient Sr_{1-x}Ca_xTiO_{3-δ}, the ‘ferroelectric’ critical concentration does not coincide with the optimal doping of superconductivity, as shown in Fig. 4 of Ref. [27].

While we are inspired by the experiment of Sr_{1-x}Ca_xTiO_{3-δ} [27], a simple analogy between doped SrTiO₃ and doped BaTiO₃ requires great care. That is because the critical concentration of doped BaTiO₂ is sufficiently high that Migdal approximation is valid. Therefore the superconductivity that we find in our calculations of doped BaTiO₃ is phonon-mediated and we can use Migdal-Eliashberg theory [1] to estimate the superconducting transition temperature. The electron-phonon coupling of doped BaTiO₃ is maximized when the polar phonon modes are most softened. This explains why the optimal doping for superconductivity in doped BaTiO₃ corresponds to the carrier density that just completely suppresses the polar distortions of BaTiO₃. This picture is supported by our first-principles calculations.

By contrast, the critical concentration of doped SrTiO₃ is so low that Migdal approximation breaks down. Compared to our results on doped BaTiO₃, the experimental fact that the “ferroelectric” critical concentration does not coincide with the optimal doping of superconductivity in Sr_{1-x}Ca_xTiO_{3-δ} implies that the superconductivity in doped SrTiO₃ is probably not purely phonon-mediated. Indeed, Ref. [27] suggests that ferroelectricity and superconductivity are two competing phases in Sr_{1-x}Ca_xTiO_{3-δ}. However, the microscopic origin of superconductivity in doped SrTiO₃ is still under intensive debate [18–25] and warrants further study.

In the revised version, we add text (highlighted in blue text) on Page 9 to mention this intriguing point.

FIG. R6: Phase diagram of doped BaTiO₃ as a function of electron concentration n . At each given electron concentration, the cubic structure C is used as the reference energy. The red triangles, green squares and blue circles correspond to rhombohedral (R), orthorhombic (O) and tetragonal (T) structures. The light blue (red) shade corresponds to the doping region in which the rhombohedral R (the tetragonal T) structure is the ground state

Response to Reviewer #3

We are very grateful to the reviewer for commenting that “while the study will be of relevance to the much-studied superconducting transition in doped SrTiO₃, it is also interesting because this case corresponds to a much higher critical carrier density” and that “the results from this study are intriguing and should motivate further investigations of the coupling between soft polar modes and itinerant electrons in doped oxides near polar phase instabilities.”

1) In Fig 1, I am wondering why the orthorhombic phase does not appear to become stabilized when increasing doping concentration. Instead the system directly changes from rhombohedral to tetragonal and cubic. Could the authors comment on that?

We thank the reviewer for this comment.

When the carrier concentration n is less than $0.1e/f.u.$, all the four crystal structures (R , O , T , C) can be stabilized during atomic relaxation by imposing symmetry. When the carrier concentration n is equal to or larger than $0.1e/f.u.$, only the cubic structure (C) can be stabilized after atomic relaxation.

However, if the carrier concentration n is less than $0.025e/f.u.$, the rhombohedral structure (R) has the lowest total energy. If the carrier concentration is larger than $0.025e/f.u.$ but smaller than $0.1e/f.u.$, the tetragonal structure (T) has the lowest total energy. To more clearly demonstrate this point, we perform additional calculations around the carrier density of $0.025e/f.u.$ and show new the results in Fig. R6.

To summarize, the orthorhombic structure of $BaTiO_3$ can be stabilized in the calculations when the symmetry is imposed and the doping is smaller than $0.1e/f.u.$. But at any doping smaller than $0.1e/f.u.$, the orthorhombic structure is not the structure with the lowest total energy in $BaTiO_3$.

In the revised version, we add a new section in the Supplementary Materials (Section VII) to more clearly show the structural transition around the carrier density of $0.025e/f.u.$.

2) I would like the authors to clarify how the electron doping impacts the p-d hybridization and Born effective charges. It seems that e-doping might screen the effective interaction between Ti-O and tune the instability. Also, it would be great if the author could also discuss the phonon behavior upon doping, especially in the rhombohedral phase.

We thank the reviewer for this comment. This comment has three sub-comments.

The first sub-comment is about $p-d$ hybridization and screening of effective metal-oxygen interactions in doped $BaTiO_3$. We first clarify that the change of $p-d$ hybridization and screening of metal-oxygen interactions are two closely related phenomena, but they are not the same thing. Previous calculations show that $p-d$ hybridization in transition metal oxides can be characterized by the occupancy of metal- d states [33, 34]. Therefore for doped $BaTiO_3$, we calculate the occupancy of Ti- d states as a function of electron dop-

FIG. R7: **a)** Illustration of the integral of Ti- d projected density of states for N_d^{val} , N_d^{cond} and N_d^{tot} (their definitions can be found in Eq. R1, R2 and R3). **b)** N_d^{val} , N_d^{cond} and N_d^{tot} for Ti- d states as a function of electron doping. Note that the left and right y -axes have different range.

ing. To provide more insights, we calculate the total occupancy of Ti- d states N_d^{tot} , the occupancy of Ti- d states in the valence bands N_d^{val} and the occupancy of Ti- d states in the conduction bands N_d^{cond} :

$$N_d^{\text{tot}} = \int_{-\infty}^{E_F} D(E) dE \quad (\text{R1})$$

$$N_d^{\text{val}} = \int_{-\infty}^{\text{in-gap}} D(E) dE \quad (\text{R2})$$

$$N_d^{\text{cond}} = \int_{\text{in-gap}}^{E_F} D(E) dE \quad (\text{R3})$$

where $D(E)$ is the density of states projected onto Ti- d orbitals. These three integrals are illustrated in Fig. R7a and their values are shown in Fig. R7b. We find that while N_d^{cond} almost linearly increases with electron doping, the total Ti- d occupancy N_d^{tot} changes much more slowly. This phenomenon is called “rehybridization effect” [35, 36]. As N_d^{cond} increases with electron doping, N_d^{val} decreases with electron doping because Ti- d and O- p states “rehybridize” in the valence states. As a result, N_d^{tot} changes slowly with electron doping. This indicates that electron doping does not significantly change the p - d hybridization in BaTiO₃.

However, it is the itinerant electrons in the conduction bands (N_d^{cond} rather than N_d^{tot})

that screen the metal-oxygen interactions in transition metal oxides, in particular long-range Coulomb interactions. Long-range Coulomb interactions favor polarization and N_d^{cond} (which linearly increases with electron doping) effectively screens the long-range Coulomb interaction, and that is why the polar displacements of BaTiO₃ are suppressed upon electron doping.

The second sub-comment is about Born effective charges. We note that with electron doping, the system becomes metallic. The Born effective charge tensor $Z_{\kappa,\gamma\alpha}^*$ of atom κ is defined as:

$$Z_{\kappa,\gamma\alpha}^* = V \frac{\partial P_\gamma}{\partial \tau_{\kappa,\alpha}} \quad (\text{R4})$$

where P_γ is the polarization and $\tau_{\kappa,\alpha}$ is the periodic displacement. However, polarization is ill-defined in metals and therefore we can not calculate the Born effective charges in doped BaTiO₃.

The third sub-comment is about the phonon behavior upon electron doping. We select three representative dopings: 0, 0.07e/f.u. and 0.14e/f.u.. In the un-doped case, the ground state structure is rhombohedral (*R*); at 0.07e/f.u. doping, the ground state structure is tetragonal (*T*) and at 0.14e/f.u. doping, the ground state structure is cubic (*C*). Fig. R8 shows the polar phonons and phonon density of states of the ground state structure at different electron dopings. Fig. R8a shows the polar phonon of the rhombohedral structure. The vibration mode of one polar phonon points along the [111] direction. The vibration mode of the other two degenerate polar phonons points perpendicular to the [111] direction. Fig. R8b shows the polar phonon of the tetragonal structure. The vibration mode of one polar phonon points along the [001] direction. The vibration mode of the other two degenerate polar phonons points to the [100] and [010] directions. Fig. R8c shows the polar phonon of the cubic structure. The vibration mode of the three degenerate polar phonon points along the [001],[010] and [100] directions. Fig. R8d, e and f show the total and atomic-projected phonon density of states of the rhombohedral structure (undoped), of the tetragonal structure (at 0.07e/f.u. doping) and of the cubic structure (at 0.14e/f.u. doping), respectively. We find that since electron doping changes the ground state crystal structure of BaTiO₃, it affects the total phonon density of states. In particular, with electron doping, the frequency of the highest optical phonon modes decreases while the low-frequency peak (around 100 cm⁻¹) increases its height.

FIG. R8: **a)** The vibration mode of the polar phonons in the rhombohedral BaTiO_3 (without doping). The vibration mode of one polar phonon points along the $[111]$ direction and the vibration mode of the other two degenerate polar phonons points perpendicular to the $[111]$ direction. **b)** The vibration mode of the polar phonons in the tetragonal BaTiO_3 (at $0.07e/f.u.$ doping). The vibration mode of one polar phonon points along the $[001]$ direction and the vibration mode of the other two degenerate polar phonons points along $[100]$ and $[010]$ directions. **c)** The vibration mode of the three degenerate polar phonons in the cubic BaTiO_3 (at $0.14e/f.u.$ doping) points along $[100]$, $[010]$ and $[001]$ directions. **d)** Phonon density of states of the rhombohedral BaTiO_3 (without doping). The blue, yellow, red and green correspond to the total, Ba-projected, Ti-projected and O-projected phonon densities of states. **e)** Phonon density of states of the tetragonal BaTiO_3 (at $0.07e/f.u.$ doping). **f)** Phonon density of states of the cubic BaTiO_3 (at $0.14e/f.u.$ doping).

In the revised version, we add two new sections in the Supplementary Materials. Section XII discusses the metal-oxygen hybridization in doped BaTiO_3 and Section XIII compares the polar phonon and phonon density of states of BaTiO_3 at different electron dopings and with different crystal symmetries.

3) I am also wondering if the author observed any localization behavior of the doped electron. In this case, added electrons (I am not sure if they are really itinerant) follows the d-band character, maybe they would also contribute to magnetism.

We thank the reviewer for this comment.

FIG. R9: **a)** The crystal structure of $2 \times 2 \times 2$ La-doped BaTiO_3 supercell with one Ba atom replaced by one La atom. **b)** An iso-value surface of conduction electrons on Ti atom in the $2 \times 2 \times 2$ La-doped BaTiO_3 supercell (for clarity only Ti atoms are shown). **c)** Total (blue) and Ti d -projected (green) density of states near the Fermi level of $2 \times 2 \times 2$ La-doped BaTiO_3 supercell. The Fermi level is shifted at zero point. **d)** The crystal structure of $3 \times 3 \times 3$ La-doped BaTiO_3 supercell with two Ti atoms replaced by two La atoms. **e)** An iso-value surface of conduction electrons on Ti atom in the $3 \times 3 \times 3$ La-doped BaTiO_3 supercell (for clarity only Ti atoms are shown). **f)** Total (blue) and Ti d -projected (green) density of states near the Fermi level of $3 \times 3 \times 3$ La-doped BaTiO_3 supercell. The Fermi level is shifted at zero point.

To study localization behavior of doped electrons, we perform additional supercell calculations of $\text{La}_x\text{Ba}_{1-x}\text{TiO}_3$. We study two representative cases: a $2 \times 2 \times 2$ supercell with one Ba atom replaced by one La atom to simulate a carrier density of $0.125e/\text{f.u.}$ (see Fig. R9a) and a $3 \times 3 \times 3$ supercell with two Ba atoms replaced by two La atoms to simulate a carrier density of $0.074e/\text{f.u.}$ (see Fig. R9d). In both cases, we show the iso-value surface of *conduction* electron density of each Ti atom and we find that the conduction electrons are uniformly distributed on Ti atoms throughout the entire simulation cell (Fig. R9b and e). The ‘dice-like’ shape of the iso-value surface indicates that the conduction electrons occupy Ti d_{xy} , d_{yz} and d_{yz} orbitals (as pointed out by the reviewer). Fig. R9c and f show the density of states of the $2 \times 2 \times 2$ supercell and $3 \times 3 \times 3$ supercell

FIG. R10: **a)** Total magnetic moment of a $2 \times 2 \times 2$ supercell $\text{La}_x\text{Ba}_{1-x}\text{TiO}_3$ ($x = 0.125$) as a function of U_{Ti} from LSDA+ U calculation. **b)** c/a ratio and Ti-O cation displacements δ of bulk BaTiO_3 as a function of U_{Ti} from LDA+ U calculation.

of $\text{La}_x\text{Ba}_{1-x}\text{TiO}_3$. The blue and green curves correspond to the total and Ti- d projected density of states. We find that there is a finite density of states around the Fermi level and it is mainly composed of Ti- d states. This indicates that the doped electrons occupy Ti- d orbitals and they are itinerant.

As for possible magnetism, we intentionally break spin symmetry and perform additional LSDA calculations. In both $2 \times 2 \times 2$ supercell and $3 \times 3 \times 3$ supercell calculations of $\text{La}_x\text{Ba}_{1-x}\text{TiO}_3$, we do not find any magnetization in our LSDA calculations. Experimentally no long-range magnetic order is observed in La doped BaTiO_3 [28, 29]. We note that if we manually increase the correlation strength on Ti d orbitals, we can artificially stabilize magnetization in doped BaTiO_3 from LSDA+ U calculations. This is due to Stoner instability [37]. Fig. R10a shows such a calculation of $2 \times 2 \times 2$ supercell of $\text{La}_x\text{Ba}_{1-x}\text{TiO}_3$ ($x = 0.125$), in which magnetization emerges when U_{Ti} is larger than 2 eV. However, it is well-known that BaTiO_3 is a band insulator with weak correlation effects [38]. More importantly, if we artificially increase the correlation strength on Ti d orbitals, we find that the polar distortions are suppressed in bulk BaTiO_3 because U_{Ti} changes the Ti-O hybridization [39]. Fig. R10b shows the c/a ratio and Ti-O displacements δ of bulk BaTiO_3 as a function of U_{Ti} . As U_{Ti} is larger than 2 eV, the polar displacements in BaTiO_3 are completely suppressed, which is at odds with the experimentally observed ferroelectric property. Therefore based on our additional calculations and known experimental results,

we conclude that long-range magnetic order (i.e. homogeneous magnetization) in doped BaTiO₃ is unlikely.

In the revised version, we add two new sections in the Supplementary Notes. Section VIII carefully studies La-doped BaTiO₃ using the supercell approach in which we demonstrate that 1) there is no localization of conduction electrons in doped BaTiO₃ and 2) conduction electrons occupy Ti-*d* states and they are itinerant. Section IX is devoted to possible magnetic properties of doped BaTiO₃ in which our calculations show that long-range magnetic order (i.e. homogeneous magnetization) in doped BaTiO₃ is unlikely.

4) The authors did not mention how the structure (lattice parameters and ionic coordinates) are changing with electron doping. Could they please provide that information?

We thank the reviewer for this comment. In the revised version, we add one new section (Sec. VII) in the Supplementary Materials in which we use four tables to list the lattice parameters and ionic coordinates of BaTiO₃ in different crystal structures at each electron doping.

We explain that as the carrier density is below $0.10e/f.u.$, all the four crystal structures (rhombohedral *R*, orthorhombic *O*, tetragonal *T*, and cubic *C*) can be stabilized. However as the carrier density is above $0.1e/f.u.$ only the cubic (*C*) structure can be stabilized. Therefore we include the structural information for *R*, *O*, *T* doped BaTiO₃ from 0 to $0.09e/f.u.$, and for *C* doped BaTiO₃ from 0 to $0.14e/f.u.$. Table S2 lists the the lattice parameters and ionic coordinates of BaTiO₃ in the rhombohedral crystal structure (*R*), Table S3 for the orthorhombic structure (*O*), Table S4 for the tetragonal structure (*T*) and Table S5 for the cubic structure (*C*).

5) The authors should try to provide a possible explanation for the large-electron phonon coupling involving the acoustic branches in the tetragonal phase, contrasting with the behavior in the cubic phase.

We thank the referee for this comment.

We provide a possible explanation, which is based on our calculations. We find that in the

FIG. R11: The vibrational mode of the acoustic phonon of doped BaTiO₃ at $\mathbf{q} = X$. **a)** at $0.09e/f.u.$ doping in the tetragonal structure. The imaginary part of electron-phonon self-energy $\text{Im}\Sigma_{\mathbf{q}\nu} = 0.39$ meV and the electron-phonon coupling $\lambda_{\mathbf{q}\nu} = 2.53$. **b)** at $0.11e/f.u.$ doping in the cubic structure. $\text{Im}\Sigma_{\mathbf{q}\nu} = 1.0 \times 10^{-4}$ meV and $\lambda_{\mathbf{q}\nu} = 1.2 \times 10^{-3}$. **c)** at $0.11e/f.u.$ doping under 0.8% compressive strain. $\text{Im}\Sigma_{\mathbf{q}\nu} = 0.75$ meV and $\lambda_{\mathbf{q}\nu} = 3.88$.

cubic structure, some electron-phonon vertices $g_{ij}^{\nu}(\mathbf{k}, \mathbf{q})$ are exactly equal to zero because some atoms are frozen in the acoustic phonons, while in the low-symmetry structure, those $g_{ij}^{\nu}(\mathbf{k}, \mathbf{q})$ become non-zero. Because $\text{Im}\Pi_{\mathbf{q}\nu} \propto |g_{ij}^{\nu}(\mathbf{k}, \mathbf{q})|^2$ [40, 41], this leads to an increase in $\text{Im}\Pi_{\mathbf{q}\nu}$. In addition, the frequencies of acoustic phonon modes $\omega_{\mathbf{q}\nu}$ are very small and $\lambda_{\mathbf{q}\nu} \propto \text{Im}\Pi_{\mathbf{q}\nu}/\omega_{\mathbf{q}\nu}^2$ [40, 41], therefore even a slight increase in $\text{Im}\Pi_{\mathbf{q}\nu}$ results in a substantial enhancement in $\lambda_{\mathbf{q}\nu}$.

To demonstrate the above point, we study a specific phonon of BaTiO₃: the acoustic phonon at $\mathbf{q} = X$. Fig. R11 compares the vibrational mode of $\mathbf{q} = X$ acoustic phonon of BaTiO₃ at $0.09e/f.u.$ doping in the tetragonal structure (**a**), at $0.11e/f.u.$ doping in the cubic structure (**b**) and at $0.11e/f.u.$ doping under 0.8% (001) compressive strain (**c**). We find that in the cubic structure (**b**), Ba atoms are strictly frozen in this acoustic phonon, while in the tetragonal structure (**a** and **c**), Ba atoms also participate in the phonon mode. Correspondingly, the imaginary part of electron-phonon self-energy $\text{Im}\Pi_{\mathbf{q}\nu}$ of doped BaTiO₃ is very small in the cubic structure, but becomes sizable in the tetragonal structure. Due to the low frequency of acoustic phonons, the electron-phonon coupling $\lambda_{\mathbf{q}\nu}$ is substantially larger in the tetragonal structure than in the cubic structure.

In the revised version, we add text on Page 10 (highlighted in blue text) to further explain this point. We also add a new section in the Supplementary Materials (Section XVI) in which we study the $\mathbf{q} = X$ acoustic phonon to demonstrate this point.

FIG. R12: Doped BaTiO₃ at an electron concentration of $0.09e/f.u.$ **a)** The schematic vibration mode of the zone-center polar optical phonon along the long c axis (z -axis) of the tetragonal structure at $0.09e/f.u.$ **b)** Comparison of density of states with and without polar phonon distortions. The red curve corresponds to the fully relaxed structure and the blue curve corresponds to the structure with the polar phonon mode imposed (the amplitude of the phonon mode is 0.12 \AA). The inset shows the near-Fermi-level density of states. **c)** The relative change of the density of states at the Fermi level $\left| \frac{D^A(E_f) - D^0(E_f)}{D^0(E_f)} \right|$ as a function of the polar phonon mode amplitude A .

6) How did the authors conclude that the phonons are mainly coupled with itinerant electrons? I am wondering if the author could compare the electronic DOS with different amplitudes of polar phonon distortions to explicitly show the coupling between itinerant electrons and the polar phonon mode.

We thank the reviewer for this good suggestion.

We study tetragonal BaTiO₃ at $0.09e/f.u.$ concentration as a representative example. Fig. R12a shows the schematic vibration mode of a polar phonon along the z -axis. Then we impose this polar phonon mode on the crystal structure with different phonon amplitudes. In Fig. R12b, we compare the density of states of doped BaTiO₃ with and without the polar phonon mode imposed (denoted by $D^A(E)$ and $D^0(E)$, respectively). Here the amplitude $A = 0.12 \text{ \AA}$. The inset of Fig. R12 clearly shows that the density of states at the Fermi level can be strongly modulated by the polar phonon. Fig. R12 shows the relative change of the density of states at the Fermi level $\left| \frac{D^A(E_f) - D^0(E_f)}{D^0(E_f)} \right|$ as a function of

phonon amplitude A . We find a substantial change in $D^A(E_f)$ as the phonon amplitude A increases. This indicates that the polar phonon is strongly coupled to itinerant electrons in doped BaTiO₃.

In the revised version, we add a pointer on Page 7 (highlighted in blue text) to refer the reader to a newly added section in the Supplementary Materials (Section XIV) in which we show that the density of states at the Fermi level can be substantially modulated by imposing polar phonon modes on the crystal structure in doped BaTiO₃.

7) In the free energy diagram at T=0 K, does the Y-axis represent the internal energy or enthalpy? In the case of complete relaxation, internal energy and enthalpy are nearly equal, while if author performed the fixed volume/structure calculation then the PV term is important.

In the free energy diagram at $T = 0$, the Y-axis represents the internal energy. In all our calculations except strain engineering (Fig. 4a in the manuscript), we perform a complete relaxation (cell parameters and internal coordinates). In the strain calculations, the in-plane lattice constants are fixed to simulate the bi-axial strain.

In the revised version, we add text on Page 12 (highlighted in blue text) to emphasize this point.

8) The discussion of potential comparison with experiments would deserve to be expanded. How robust do the authors believe their conclusions are if one is to include explicit chemical disorder (dopants or vacancies)? The authors mention that comparison with electrostatic doping experiments could be easier or more direct, and point out (Ref 53) the reported observation of superconductivity in KTaO₃ upon electrostatic carrier doping. If so, why not perform/extend the theory to the case of KTaO₃, which could already afford a direct test?

We thank the reviewer for this comment and the suggestion of crosschecking KTaO₃.

We agree with the reviewer that chemical disorder may affect the transport properties of the normal state (the metallic state above the superconducting transition temperature).

FIG. R13: **a)** The electron-phonon spectral function $\alpha^2 F(\omega)$ of BaTiO₃ at 0.09e/f.u. doping (top) and of KTaO₃ at 0.14e/f.u. doping (bottom). The dashed lines are the accumulative electron-phonon coupling. **b)** The superconducting transition temperature T_c of BaTiO₃ at 0.09e/f.u. doping (red) and of KTaO₃ at 0.14e/f.u. doping (blue) that is estimated by Eliashberg equation as a function of Morel-Anderson pseudopotential μ_{ij}^* .

However, according to Anderson's theorem, superconductivity in a conventional superconductor is robust with respect to non-magnetic disorder in the host material [31]. The superconducting transition temperature T_c of a conventional superconductor barely depends on material purity. In our case, the superconductivity in doped BaTiO₃ is phonon-mediated (i.e. conventional) and La is a non-magnetic dopant. Therefore Anderson's theorem applies and we expect that the superconducting phase in doped BaTiO₃ is robust against chemical disorder.

We also perform new calculations of electron-phonon coupling for KTaO₃ at a representative electron concentration of 0.14e/f.u. (based on the experiment [30]). Fig. R13a compares the electron-phonon spectral function $\alpha^2 F(\omega)$ between doped KTaO₃ (at 0.14e/f.u.) and doped BaTiO₃ (at 0.09e/f.u.). We find that while the electron-phonon coupling of doped BaTiO₃ at 0.09e/f.u. is 0.61, doped KTaO₃ at 0.14e/f.u. has a smaller electron-phonon coupling of 0.36. Fig. R13b compares the superconducting transition temperature that is estimated by the Eliashberg equation. We find that since the electron-phonon coupling of KTaO₃ at 0.14e/f.u. doping is smaller than that of BaTiO₃ at 0.09e/f.u. doping, the KTaO₃ superconducting transition temperature is also smaller than the doped BaTiO₃, given that the two materials have similar Morel-Anderson pseudopotential μ^* .

It is very difficult (almost impossible) to use Eliashberg equation to determine a very small superconducting transition temperature (such as 50 mK for KTaO_3 at $0.14e/f.u.$ doping). Our calculations of doped KTaO_3 find that if $\mu_{ij}^* \geq 0.1$, the superconducting transition temperature of KTaO_3 (at $0.14e/f.u.$ doping) is not higher than 0.1 K, which is in reasonable agreement with the experiment [30].

In the revised version, we expand our discussion of potential experiments on Page 11 and 12 (highlighted in blue text). We also add a section in the Supplementary Materials (Section XV) to show the comparison between doped BaTiO_3 and doped KTaO_3 .

Response to Reviewer #4

We thank the reviewer for the comment “I find the paper ‘A large modulation of electron-phonon coupling and an emergent superconducting dome in doped strong ferroelectrics’ to be timely and potentially relevant for generation of interest in the ferroelectrics as a playground for finding possible new superconductors.”

1) “previous studies found that in n-doped BaTiO_3 , increasing the carrier density gradually reduces its polar distortions and induces a continuous polar-to-centrosymmetric phase transition (similar to $\text{Sr}_{1-x}\text{Ca}_x\text{TiO}_{3-\delta}$)”. Is the picture for the phase transition in Ca-STO that simple? I think it is a bit more involved than that.

We thank the reviewer for this helpful comment.

We agree with the reviewer that the quantum phase transition in $\text{Sr}_{1-x}\text{Ca}_x\text{TiO}_{3-\delta}$ [27] is more complicated than what is described for n -doped BaTiO_3 . The reason that we mention $\text{Sr}_{1-x}\text{Ca}_x\text{TiO}_{3-\delta}$ in the introduction is to provide another relevant experimental system, which inspires us to study doped BaTiO_3 . However, as the reviewer commented below (comment #19), a direct comparison between n -doped BaTiO_3 and $\text{Sr}_{1-x}\text{Ca}_x\text{TiO}_{3-\delta}$ is premature and therefore we remove the phrase “(similar to $\text{Sr}_{1-x}\text{Ca}_x\text{TiO}_{3-\delta}$)” on Page 3 in the revised version.

2) **“the critical concentration for the phase transition is about $10^{21}/\text{cm}^3$, which is high enough so that the electron-phonon coupling can be directly calculated within the Migdals approximation”**. I dont think the comparison between the Ca case and other cases is that straightforward. Also, this seems like cherry picking in terms of very large doping range in STO for which it superconducts.

With respect, we are a little confused by the correlation between the quote and the reviewer’s comment here. The quote “the critical concentration for the phase transition is about $10^{21}/\text{cm}^3$...” refers to doped BaTiO_3 (not SrTiO_3). That statement says that the critical concentration for doped BaTiO_3 is high enough so that the Migdal’s approximation is valid. However, the comment from the reviewer seems to be concerning doped SrTiO_3 . While we agree with the reviewer that different doping mechanisms in SrTiO_3 (La-doping, Nb-doping, oxygen-vacancies etc.) may have some difference in the superconducting properties of SrTiO_3 , the quote is not directly related to doped SrTiO_3 .

If we misunderstood the reviewer’s comment, we kindly request the reviewer to further clarify it and we are happy to address it again.

3) **“Motivated by the above experiments and theories”**. The authors should cite other experimental works on STO, some a listed here M. N. Gastiasoro et al., Superconductivity in Dilute SrTiO_3 : A Review, Ann. Phys. 168107 (2020)

We thank the reviewer for bringing this newly-published comprehensive review to our attention, which expands our knowledge on doped SrTiO_3 . We cite it and other experiment works on SrTiO_3 in our revised version.

In the revised version, we cite the following papers Ann. Phys. 168107 (2020) (now Reference 38), Phys. Rev. Materials **3**, 124801 (2019) (now Reference 23), Scientific Reports **6**, 37582 (2016) (now Reference 24).

4) **“superconductivity emerges at a much lower carrier concentration $10^{17} - 10^{20}/\text{cm}^3$, which invalidates the Migdals approximation and Eliashberg equation”**. The argument should be spelled out more precisely in terms of

the energy scales that are being compared with each other. And what are the actual experimental magnitudes for those compared values? Some accuracy wont hurt here.

We thank the reviewer for this suggestion. The energy scale we compare here is the Debye frequency ($\hbar\omega_D$) versus Fermi energy (ϵ_F). For the Migdal's approximation to be valid, the Debye frequency is usually much smaller than Fermi energy ($\hbar\omega_D/\epsilon_F \sim 10^{-2} - 10^{-3}$). However for doped SrTiO₃, since the carrier concentration is very low, the Fermi energy varies between 2 and 60 meV. On the other hand, the role of Debye frequency ω_D can be replaced by the longitudinal optical phonon frequency ω_L , which is on the order of 100 meV for SrTiO₃ [42]. Therefore in doped SrTiO₃, $\hbar\omega_D/\epsilon_F \sim 1 - 10^2$, which violates the Migdal criterion.

In the revised version, we add text on Page 3 (highlighted in blue text) to make this point more clear.

5) “we find that the phonon bands associated with the soft polar optical phonons are strongly coupled to itinerant electrons across the polar-to-centrosymmetric phase transition in doped BaTiO₃”. Whats the insight for “why” this is the case?

We thank the referee for this important comment.

For a canonical polar metal such as LiOsO₃, across the polar-to-centrosymmetric phase transition, the coupling between the soft polar optical phonons and itinerant electrons is very weak [43]. This is known as the ‘weak coupling’ mechanism [43, 44]. The weak coupling in LiOsO₃ is due to the fact that the itinerant electrons reside on Os-*d* orbitals while the polar phonons involve the movement of Li and O [43]. However, in doped BaTiO₃, itinerant electrons reside on Ti-*d* orbitals (see Fig. 2a in the main text) and the polar phonons mainly involve Ti and O movement (see Fig. 2c in the main text). Because the itinerant electrons and polar phonon are associated with the same atoms in doped BaTiO₃, the coupling is strong, while in LiOsO₃ the itinerant electrons and polar phonon involve different atoms and thus the coupling is weak.

In the revised version, we add text on Page 7 (highlighted in blue text) to further clarify this point.

6) “In addition, we find that close to the critical concentration, lowering the crystal symmetry of doped BaTiO₃ by imposing epitaxial strain further increases the superconducting temperature via a sizable coupling between itinerant electrons and acoustic phonon bands.”. What about the ferroelectric phonon to electron coupling change with strain?

We thank the reviewer for this important comment. Since this comment is related to a few other comments (comments #10, #11, #12, #15), we would like to introduce a normalized around-zone-center branch-resolved electron-phonon coupling λ_ν as:

$$\lambda_\nu = \frac{\int_{|\mathbf{q}| < q_c} d\mathbf{q} \lambda_{\mathbf{q}\nu}}{\int_{|\mathbf{q}| < q_c} d\mathbf{q}} \quad (\text{R5})$$

where q_c is a small number within which there are no phonon band crossings. There are two reasons to introduce this definition: 1) exactly at the zone-center Γ point, the acoustic phonon frequency is zero. Since $\lambda_{\mathbf{q}\nu} \propto 1/\omega_{\mathbf{q}\nu}$, the contribution from the acoustic mode is ill-defined at Γ point. 2) Because there are no phonon band crossings within $|\mathbf{q}| < q_c$, each phonon mode (labelled by ν) is well-defined. For a general \mathbf{q} point, it is not trivial to distinguish which phonon band corresponds to polar modes and which to other optical modes. We choose $q_c = 0.05 \frac{\pi}{a}$ where a is the lattice constant. The qualitative conclusions do not depend on the choice of q_c (as long as no phonon band crossings occur within $|\mathbf{q}| < q_c$).

Now we apply the above definition of λ_ν to cubic BaTiO₃ at 0.11e/f.u. doping. Close to Γ point, the three lowest phonon bands ($\nu = 1 - 3$) are acoustic modes, while the next three lowest phonon bands ($\nu = 4 - 6$) are polar modes (this is not true for a general \mathbf{q} -point in the Brillouin zone). We find that without epitaxial strain:

$$\begin{aligned} \lambda_{\text{acoustic}} &= \sum_{\nu=1-3} \lambda_\nu = 0.27 \\ \lambda_{\text{polar}} &= \sum_{\nu=4-6} \lambda_\nu = 5.58 \end{aligned} \quad (\text{R6})$$

Under 0.8% (001) compressive strain, we find:

$$\begin{aligned}\lambda_{\text{acoustic}} &= \sum_{\nu=1-3} \lambda_{\nu} = 4.45 \\ \lambda_{\text{polar}} &= \sum_{\nu=4-6} \lambda_{\nu} = 3.21\end{aligned}\tag{R7}$$

Therefore under 0.8% (001) compressive strain, λ_{polar} remains substantial (albeit reduced by 40%), but $\lambda_{\text{acoustic}}$ is increased by one order of magnitude.

With these new numbers of λ_{ν} , we revise our text and make a more quantitative comparison on Page 6, 7 and 10 (highlighted in blue text).

7) “Our results show that the weakly coupled electron mechanism in ‘ferroelectric-like metals’ is not necessarily present in doped strong ferroelectrics and as a consequence, the soft polar phonons can be utilized to induce phonon-mediated superconductivity across a structural phase transition.”. This needs more explanation for why “is not necessary present. Right now, its just a statement but not an explanation, which would be helpful to have here or to come back to this point later in the text to explain “why”.

We thank the referee for this comment. We explain this in our reply to the comment #5.

In the revised version, we add text on Page 7 (highlighted in blue text) to explain this point.

8) “Fig. 1a shows that as electron doping concentration n increases from 0 to $0.15e/f.u.$, $BaTiO_3$ transitions from the rhombohedral structure to the tetragonal structure, and finally to the cubic structure.”. Is the role of disorder due to dopants considered or derived here? How is its relevance in real materials ruled out? Any experimental study that could be cited?

We thank the referee for this comment.

In our calculations of Fig. 1, we do not explicitly consider disorder due to dopants. We assume that the crystal structure of $BaTiO_3$ is homogeneous upon chemical doping and manually change the number of electrons to simulate the doping effects (similar to the

previous work [9]). Our assumption is supported by the experiments that bulk La-doped BaTiO₃ were synthesized in single phase ceramics with homogeneous microstructures [45] and high-quality La-doped BaTiO₃ thin films were grown and exhibit metallic behaviors [28, 29, 46].

More importantly, we would like to note that while chemical disorder may affect transport properties of normal states (the metallic state above superconducting transition temperatures), Anderson's theorem asserts that superconductivity in a conventional superconductor is robust with respect to non-magnetic disorder in the host material [31]. That is, the superconducting transition temperature T_c of a conventional superconductor barely depends on material purity. In our case, the superconductivity in doped BaTiO₃ is phonon-mediated (i.e. conventional) and La is a non-magnetic dopant. Therefore Anderson's theorem applies and we expect that even if chemical disorder may arise in actual experiment, it does not affect the superconducting properties of doped BaTiO₃.

In the revised version, we add text to explain possible effects from chemical disorder in the Discussion Section (on Page 11, highlighted in blue text).

9) “Furthermore the low electron concentration in the rhombohedral structure invalidates Migdals theorem and electron-phonon coupling can not be calculated within Migdals approximation.”. The authors should probably spell out which values compared to which to support this statement, not clear. (It is in the SM but some readers may not get there, or provide a pointer).

We thank the reviewer for pointing this out.

The values compared here are the Debye frequency ($\hbar\omega_D$) and the Fermi energy (ϵ_F), which can be converted to the Debye temperature ($T_D = \hbar\omega_D/k_B$) and the Fermi temperature ($T_F = \epsilon_F/k_B$). If T_D/T_F is small, then Migdal's approximation is valid. The Fermi energy (or the Fermi temperature) is correlated with the electron concentration. When the electron concentration is sufficiently low, the Fermi temperature can be comparable to or even lower than the Debye temperature, which invalidates Migdal's approximation.

In the revised version, we add text on Page 3 (highlighted in blue text) to explicitly

explain that the Debye frequency of doped SrTiO₃ is comparable to or even higher than the Fermi temperature and hence the Migdal’s approximation breaks down. We also add a pointer on Page 4 (highlighted in blue text) that refers the readers to the Section V: “Validation test of Migdal’s theorem” in the Supplementary Materials (as suggested by the reviewer).

10) “that inversion symmetry breaking by collective polar displacements in a metal relies on a weak coupling between itinerant electrons and soft phonons responsible for removing inversion symmetry.” I know that this is being cited in the literature as weak vs. strong coupling regularly, but it would be nice to quantify the statements like this, what is weak and what is strong limit? Can we define for the readers benefit and also for the accuracy of the statements?

We thank the reviewer for this important comment.

Following our reply to the comment #6, we use the branch-resolved electron-phonon coupling λ_ν to quantify the coupling strength. For doped BaTiO₃ at 0.09e/f.u. doping, we find:

$$\lambda_{\text{polar}} = \sum_{\nu=4-6} \lambda_\nu = 10.92 \quad (\text{R8})$$

and at 0.11e/f.u. doping, we find:

$$\lambda_{\text{polar}} = \sum_{\nu=4-6} \lambda_\nu = 5.58 \quad (\text{R9})$$

On the other hand, the canonical polar metal LiOsO₃ is known to have a weak coupling between itinerant electrons and polar phonons [43]. Using our definition of branch-resolved electron-phonon coupling, we find that for LiOsO₃ (the zone-center polar modes of LiOsO₃ are $\nu = 5, 6, 9$),

$$\lambda_{\text{polar}} = \sum_{\nu=5,6,9} \lambda_\nu = 0.50 \quad (\text{R10})$$

Thus λ_{polar} of doped BaTiO₃ is one order of magnitude larger than that of LiOsO₃.

Theoretically, $\lambda_{\text{polar}} \geq 0$ and there is no upper limit. Usually if $\lambda_{\text{polar}} > 1$, we may consider the coupling as strong (of course, this is not a strict criterion).

In the revised version, we add text on Page 6 and 7 (highlighted in blue text) to make a quantitative statement of “weak” and “strong” couplings.

FIG. R14: Branch-resolved around-zone-center electron-phonon coupling λ_ν of BaTiO₃ at **a)** 0.09e/f.u. doping in the tetragonal (*T*) structure and **b)** 0.11e/f.u. doping in the cubic (*C*) structure. The definition of λ_ν is given in Eq. (R5) with $q_c = 0.05\frac{\pi}{a}$ and a is the lattice constant. The red bars correspond to acoustic phonons, the blue bar correspond to polar phonons and the green bars correspond to other optical phonons.

11) “the phonon bands associated with the zone-center polar phonons have the strongest coupling to itinerant electrons, while the couplings of other phonon bands are much weaker.”. This sounds like a relative arguments, can it be made more precise here in terms of the actual numbers?

We thank the reviewer for this good suggestion. Following our reply to the comment #6, we calculate each λ_ν of BaTiO₃ at 0.09e/f.u. doping and at 0.11e/f.u. doping. The results are shown in Fig. R14. We find that for 0.09e/f.u. doping,

$$\begin{aligned}
 \lambda_{\text{acoustic}} &= \sum_{\nu=1-3} \lambda_\nu = 3.83 \\
 \lambda_{\text{polar}} &= \sum_{\nu=4-6} \lambda_\nu = 10.92 \\
 \lambda_{\text{others}} &= \sum_{\nu=7-15} \lambda_\nu = 0.53
 \end{aligned}
 \tag{R11}$$

and for $0.11e/f.u.$ doping,

$$\begin{aligned}\lambda_{\text{acoustic}} &= \sum_{\nu=1-3} \lambda_{\nu} = 0.27 \\ \lambda_{\text{polar}} &= \sum_{\nu=4-6} \lambda_{\nu} = 5.58 \\ \lambda_{\text{others}} &= \sum_{\nu=7-15} \lambda_{\nu} = 0.11\end{aligned}\tag{R12}$$

In both cases, λ_{polar} is larger than $\lambda_{\text{acoustic}}$ and λ_{others} .

With these new numbers of λ_{ν} , we revise our text and make a more quantitative comparison on Page 6 and 7 (highlighted in blue text).

12) “and therefore can also make non-negligible contribution to the total electron-phonon coupling”. Provide a number, its hard to judge the acoustic phonons contribution based on the plots in Fig 3. I cannot really see the amplification effect. Can it be also made more quantitative statement?

We thank the referee for this comment and we apologize for the confusion that is caused by the phrase “amplification effect”.

We first clarify that the mode-resolved electron-phonon coupling $\lambda_{\mathbf{q}\nu}$ is [40, 41]:

$$\lambda_{\mathbf{q}\nu} = \frac{1}{\pi N_F} \frac{\text{Im}\Pi_{\mathbf{q}\nu}}{\omega_{\mathbf{q}\nu}^2} \quad \text{and} \quad \text{Im}\Pi_{\mathbf{q}\nu} = \pi\omega_{\mathbf{q}\nu} \sum_{ij} \int \frac{d\mathbf{k}}{V_{\text{BZ}}} |g_{ij}^{\nu}(\mathbf{k}, \mathbf{q})|^2 \delta(\epsilon_{j\mathbf{k}} - \epsilon_F) \delta(\epsilon_{i\mathbf{k}+\mathbf{q}} - \epsilon_F)\tag{R13}$$

For a low phonon frequency $\omega_{\mathbf{q}\nu}$ such as those of acoustic phonon modes, a slight increase in the electron-phonon self-energy $\text{Im}\Pi_{\mathbf{q}\nu}$ will result in a substantial increase in $\lambda_{\mathbf{q}\nu}$ because $\omega_{\mathbf{q}\nu}$ is in the denominator. This is what we mean by “amplification effect”. Nevertheless, we appreciate the reviewer’s confusion and in the revised version, we modify the text on Page 9 (highlighted in blue text) and make this point more clear.

Second, following the comment #6 and #11, we now have well-defined numbers for mode-resolved electron-phonon couplings. Within the region $|\mathbf{q}| < q_c$ in which no phonon crossing occurs and each phonon branch can be well assigned to acoustic and polar modes,

we find that

$$\begin{aligned} \text{at } 0.09e/\text{f.u. doping in the } T \text{ structure} \quad & \frac{\lambda_{\text{acoustic}}}{\lambda_{\text{acoustic}} + \lambda_{\text{polar}} + \lambda_{\text{others}}} = 25\% \quad (\text{R14}) \\ \text{at } 0.11e/\text{f.u. doping in the } C \text{ structure} \quad & \frac{\lambda_{\text{acoustic}}}{\lambda_{\text{acoustic}} + \lambda_{\text{polar}} + \lambda_{\text{others}}} = 5\% \end{aligned}$$

This quantitatively shows that around the zone-center Γ point, the acoustic modes make non-negligible contribution to the electron-phonon coupling (the acoustic mode contribution is larger in the tetragonal T structure than in the cubic C structure).

In the revised version, we modify the paragraph about acoustic phonons on Page 9 and 10 (highlighted in blue text) and make more quantitative statements. We also revise Fig. 4c which demonstrates the ‘‘amplification effect’’ of $\lambda_{\mathbf{q}\nu}$ in the acoustic modes of doped BaTiO₃ under (001) compressive strain.

13) ‘‘a thought-experiment’’ should probably be ‘‘a numerical experiment’’ in several places.

We thank the reviewer for this suggestion. We replace ‘‘a thought-experiment’’ with ‘‘a numerical experiment’’ on Page 9 and 10 in the revised version.

14) ‘‘ λ increases from 0.50 in the P m3m structure to 0.57 in the P 4mm structure, and the superconducting transition temperature T_c increases from 0.76 K in the P m3m structure to 2.0 K in the P 4mm structure’’. I dont know if we can trust Eliashberg equation and McMillans formula with such small change in alfa and draw conclusions that 2 K is 2 K, it might be 2 might be 10 in real material. Id like to see some comment about the accuracy of the expectation. These predictions are quite often off by a numerical factor.

We thank the reviewer for this insightful comment.

We agree with the reviewer that there is uncertainty in Eliashberg equations and McMillan’s formula, such as the Morel-Anderson pseudopotential μ^* which is treated as a parameter. Since the superconducting transition temperature T_c strongly depends on μ^* (see Fig. 3f in the main text), we only use Eliashberg equations and McMillan’s formula to

estimate T_c . The take-home message here is that when epitaxial strain lowers the crystal symmetry of doped BaTiO₃, its electron-phonon coupling λ is increased, which leads to an increase in T_c .

Nevertheless, we appreciate the reviewer's concern and in the revised version we remove the estimated superconducting transition temperatures on Page 9.

15) “The difference in $\text{Im}\Pi_{\mathbf{q}\nu}$ is further amplified by the low phonon frequencies $\omega_{\mathbf{q}\nu}$. Fig. 4c explicitly compares the mode-resolved electron-phonon coupling $\lambda_{\mathbf{q}\nu}$ for the lowest phonon band of the two doped BaTiO₃. It is evident that $\lambda_{\mathbf{q}\nu}$ is substantially larger in 0.11e/f.u. concentration (space group Pm3m), and then we impose a slight compressive the optimized lattice constant a is 3.972 Å; under a 0.8% biaxial compressive strain, the ground state structure becomes tetragonal with the optimized short lattice constant a being electron-phonon coupling λ increases from 0.50 in the P m3m structure to 0.57 in the P 4mm structure, and the superconducting transition temperature T_c increases from 0.76 K in the P 4mm tetragonal structure than in the P m3m cubic structure.”. I'd like to see the mode-resolved coupling for the FE modes next to the acoustic ones. How is the FE one changed by strain? What is the scale of this mode-resolved lambda? Is it not the same as that of the averaged one?

We thank the reviewer for this comment.

Please refer to our reply to the comment #6 in which we define λ_{ν} for acoustic and polar modes, and we compare λ_{ν} in the absence of strain versus in the presence of strain.

We note that the around-zone-center branch-resolved electron-phonon coupling λ_{ν} can be substantially larger than the total electron-phonon coupling λ . That is because 1) we normalize λ_{ν} (we divide by $\int_{|\mathbf{q}|<q_c} d\mathbf{q}$ to make λ_{ν} dimensionless) and 2) $\lambda_{\mathbf{q}\nu}$ is substantial when \mathbf{q} is around the high-symmetry cuts (e.g. $\Gamma \rightarrow X$) and is negligible in other regions of the phonon Brillouin zone.

Following our reply to the comment #12, we modify the paragraph of acoustic phonons

and make more quantitative discussions on Page 9 and 10 (highlighted in blue text). We also revise Fig. 4c in which we show the mode-resolved electron-phonon coupling $\lambda_{\mathbf{q}\nu}$ along Γ to X .

16) “the exactly same electron concentration, indicating that the additional increase in $\text{Im}\Pi_{\mathbf{q}\nu}$ arises solely from the crystal structure difference. Lowering the symmetry of BaTiO₃ crystal structure allows more electron-phonon scattering processes that would be forbidden in the cubic structure by symmetry considerations.”. Is this a quantitative or qualitative statement? I suggest quoting numbers. Is this general or specific to these calculations? Seems to be too important to pass through so quickly - need more clarification and quantitative argumentation beyond the plots show. Is there a more general insight to be learned?

We thank the reviewer for this comment.

Following our reply to the comments #6, we find that for BaTiO₃ at 0.11e/f.u. doping:

$$\begin{aligned} \text{without epitaxial strain} \quad \lambda_{\text{acoustic}} &= 0.27 & (\text{R15}) \\ \text{under 0.8\% (001) compressive strain} \quad \lambda_{\text{acoustic}} &= 4.45 \end{aligned}$$

This result is specific to doped BaTiO₃. A possible explanation, which is based on our calculations, is that in the cubic structure, some electron-phonon vertices $g'_{ij}(\mathbf{k}, \mathbf{q})$ are exactly equal to zero because some atoms are frozen in the acoustic phonons, while in the low-symmetry structure, those $g'_{ij}(\mathbf{k}, \mathbf{q})$ become non-zero. Because $\text{Im}\Pi_{\mathbf{q}\nu} \propto |g'_{ij}(\mathbf{k}, \mathbf{q})|^2$ (see Eq. R13), this leads to an increase in $\text{Im}\Pi_{\mathbf{q}\nu}$. In addition, the frequencies of acoustic phonon modes $\omega_{\mathbf{q}\nu}$ are very small and $\lambda_{\mathbf{q}\nu} \propto \text{Im}\Pi_{\mathbf{q}\nu}/\omega_{\mathbf{q}\nu}^2$ (see Eq. R13), therefore even a slight increase in $\text{Im}\Pi_{\mathbf{q}\nu}$ results in a more substantial enhancement in $\lambda_{\mathbf{q}\nu}$.

In the revised version, we modify text on Page 10 (highlighted in blue text) to further explain this point. We also add a new section in the Supplementary Materials (Section XVI) in which we study a specific acoustic phonon to demonstrate this point.

17) “a small compressive strain that lowers the crystal symmetry can also increase its superconducting transition temperature, similar to doped SrTiO₃”.

Please specify compressive strain in which direction. Also please see other relevant experiments in ref. M. N. Gastiasoro, Superconductivity in Dilute SrTiO₃: A Review, Ann. Phys. 168107 (2020). In terms of the polar modes and anisotropies the strain direction is important.

We thank the reviewer for this comment and we apologize for this omission.

As Fig. 4 in the main text shows, the epitaxial strain is bi-axial, i.e. the in-plane lattice constants of doped BaTiO₃ (a and b) are either compressed or elongated so as to match the lattice constant of the substrate. The out-of-plane lattice constant of doped BaTiO₃ (c) increases (decreases) when the bi-axial strain is compressive (tensile). This type of strain is known as (001) compressive strain.

In the revised version, we add text on Page 9 and 10 (highlighted in blue text) to explicitly explain that the bi-axial (001) compressive strain is imposed such that the in-plane two lattice constants (a and b) are fixed to a smaller value. This (001) compressive strain can increase the electron-phonon coupling of doped BaTiO₃ and possibly superconducting transition temperature.

18) “Since our simulation does not consider dopants explicitly, a more desirable doping method is to use electrostatic carrier doping”. But then the structure doesn’t not change. So how does electrostatic gating of the surface should work in terms of the structural phase transition? Are the authors saying that just strain and surface conductivity enough? Please clarify with more precision.

We thank the reviewer for this comment.

When we discuss electrostatic gating or epitaxial strain, we mean that those methods should be applied to BaTiO₃ thin films (not bulk). Experiments show that electrostatic gating can induce electrons into a region of KTaO₃ that is a few nanometers thick from the surface [30]. This leads to an effective three-dimensional carrier density of $2.2 \times 10^{21} \text{cm}^{-3}$ (equivalently $0.14e/\text{f.u.}$) and thus superconductivity of about 50 mK in doped KTaO₃.

We expect that in BaTiO₃ thin films (a few nanometers thick), electrostatic method

or epitaxial strain can modulate the structural properties of the entire film [30, 47, 48]. Therefore based on our calculations, when the electron-phonon coupling of doped BaTiO₃ is sufficiently enhanced around the critical concentration, conventional superconductivity may emerge in BaTiO₃ thin films.

In the revised version, we add text on Page 11 (highlighted in blue text) to discuss a number of new details about the electrostatic doping method.

19) “We hope that our predictions will stimulate new experiments on doped ferroelectrics and, if confirmed, may help shed light on the mysterious origin of the superconductivity in doped SrTiO₃.” I dont think the projection for STO is that simple, given that the authors say that for their material Migdals approximation is valid, while for STO it is not that straightforward. I think the overall projected connection to STO is somewhat premature, many more details would need to be covered to make any such conclusion.

We thank the reviewer for this comment and we agree with the reviewer that a direction connection between superconductivity in doped BaTiO₃ and that in doped SrTiO₃ is premature. In the revised version, we remove the sentence “if confirmed, may help shed light on the mysterious origin of the superconductivity in doped SrTiO₃.” on Page 12.

-
- [1] A. Migdal, Soviet Physics Journal of Experimental and Theoretical Physics **7**, 996 (1958).
 - [2] Y. Yao, J. S. Tse, Y. Ma, and K. Tanaka, Europhysics Letters **78**, 37003 (2007).
 - [3] M. I. Eremets, I. A. Trojan, S. A. Medvedev, J. S. Tse, and Y. Yao, Science **319**, 1506 (2008).
 - [4] D. Duan, Y. Liu, F. Tian, D. Li, X. Huang, Z. Zhao, H. Yu, B. Liu, W. Tian, and T. Cui, Scientific Reports **4**, 6968 (2014).
 - [5] A. P. Drozdov, M. I. Eremets, I. A. Troyan, V. Ksenofontov, and S. I. Shylin, Nature **525**, 73 (2015).
 - [6] H. Liu, I. I. Naumov, Z. M. Geballe, M. Somayazulu, J. S. Tse, and R. J. Hemley, Phys. Rev. B **98**, 100102 (2018).

- [7] A. P. Drozdov, P. P. Kong, V. S. Minkov, S. P. Besedin, M. A. Kuzovnikov, S. Mozaffari, L. Balicas, F. F. Balakirev, D. E. Graf, V. B. Prakapenka, et al., *Nature* **569**, 528 (2019).
- [8] C. Xia, Y. Chen, and H. Chen, *Phys. Rev. Materials* **3**, 054405 (2019).
- [9] Y. Wang, X. Liu, J. D. Burton, S. S. Jaswal, and E. Y. Tsymbal, *Phys. Rev. Lett.* **109**, 247601 (2012).
- [10] E. Heifets, E. Kotomin, and V. A. Trepakov, *Journal of Physics: Condensed Matter* **18**, 4845 (2006).
- [11] L. Rimai and G. A. deMars, *Phys. Rev.* **127**, 702 (1962).
- [12] J. G. Bednorz and K. A. Müller, *Phys. Rev. Lett.* **52**, 2289 (1984).
- [13] P. Ghosez, E. Cockayne, U. V. Waghmare, and K. M. Rabe, *Phys. Rev. B* **60**, 836 (1999).
- [14] D. I. Bilc, R. Orlando, R. Shaltaf, G.-M. Rignanesi, J. Íñiguez, and P. Ghosez, *Phys. Rev. B* **77**, 165107 (2008).
- [15] Y. Zhang, J. Sun, J. P. Perdew, and X. Wu, *Phys. Rev. B* **96**, 035143 (2017).
- [16] X. Lin, G. Bridoux, A. Gourgout, G. Seyfarth, S. Krämer, M. Nardone, B. Fauqué, and K. Behnia, *Phys. Rev. Lett.* **112**, 207002 (2014).
- [17] X. Lin, Z. Zhu, B. Fauqué, and K. Behnia, *Phys. Rev. X* **3**, 021002 (2013).
- [18] V. Kozii, Z. Bi, and J. Ruhman, *Phys. Rev. X* **9**, 031046 (2019).
- [19] K. Dunnett, A. Narayan, N. A. Spaldin, and A. V. Balatsky, *Phys. Rev. B* **97**, 144506 (2018).
- [20] J. M. Edge, Y. Kedem, U. Aschauer, N. A. Spaldin, and A. V. Balatsky, *Phys. Rev. Lett.* **115**, 247002 (2015).
- [21] P. Wölfle and A. V. Balatsky, *Phys. Rev. B* **98**, 104505 (2018).
- [22] J. Ruhman and P. A. Lee, *Phys. Rev. B* **94**, 224515 (2016).
- [23] L. P. Gor'kov, *Proceedings of the National Academy of Sciences* **113**, 4646 (2016).
- [24] S. Kanasugi and Y. Yanase, *Phys. Rev. B* **100**, 094504 (2019).
- [25] D. van der Marel, F. Barantani, and C. W. Rischau, *Phys. Rev. Research* **1**, 013003 (2019).
- [26] Y. Tomioka, N. Shirakawa, K. Shibuya, and I. H. Inoue, *Nature Communications* **10**, 738 (2019).
- [27] C. W. Rischau, X. Lin, C. P. Grams, D. Finck, S. Harms, J. Engelmayer, T. Lorenz, Y. Gallais, B. Fauqu, J. Hemberger, et al., *Nat. Phys.* **13**, 643 (2017).
- [28] K. S. Takahashi, Y. Matsubara, M. S. Bahramy, N. Ogawa, D. Hashizume, Y. Tokura, and

- M. Kawasaki, *Scientific Reports* **7** (2017).
- [29] A. Zhang, Q. Li, D. Gao, M. Guo, J. Feng, Z. Fan, D. Chen, M. Zeng, X. Gao, G. Zhou, et al., *Journal of Physics D: Applied Physics* **53**, 025301 (2019).
- [30] K. Ueno, S. Nakamura, H. Shimotani, H. T. Yuan, N. Kimura, T. Nojima, H. Aoki, Y. Iwasa, and M. Kawasaki, *Nature Nanotechnology* **6**, 408 (2011).
- [31] P. Anderson, *Journal of Physics and Chemistry of Solids* **11**, 26 (1959).
- [32] H. Yuan, H. Shimotani, A. Tsukazaki, A. Ohtomo, M. Kawasaki, and Y. Iwasa, *Advanced Functional Materials* **19**, 1046 (2009).
- [33] X. Wang, M. J. Han, L. de' Medici, H. Park, C. A. Marianetti, and A. J. Millis, *Phys. Rev. B* **86**, 195136 (2012).
- [34] H. Park, A. J. Millis, and C. A. Marianetti, *Phys. Rev. B* **90**, 235103 (2014).
- [35] C. A. Marianetti, G. Kotliar, and G. Ceder, *Phys. Rev. Lett.* **92**, 196405 (2004).
- [36] M. N. Grisolia, J. Varignon, G. Sanchez-Santolino, A. Arora, S. Valencia, M. Varela, R. Abrudan, E. Weschke, E. Schierle, J. E. Rault, et al., *Nature Physics* **12**, 484 (2016).
- [37] E. C. Stoner, *Proceedings of the Royal Society of London. Series A. Mathematical and Physical Sciences* **165**, 372 (1938).
- [38] T. Kolodiaznyy, M. Tachibana, H. Kawaji, J. Hwang, and E. Takayama-Muromachi, *Phys. Rev. Lett.* **104**, 147602 (2010).
- [39] N. A. Benedek and T. Birol, *J. Mater. Chem. C* **4**, 4000 (2016).
- [40] E. R. Margine and F. Giustino, *Phys. Rev. B* **87**, 024505 (2013).
- [41] F. Giustino, *Rev. Mod. Phys.* **89**, 015003 (2017).
- [42] M. N. Gastiasoro, J. Ruhman, and R. M. Fernandes, *Annals of Physics* **417**, 168107 (2020).
- [43] N. J. Laurita, A. Ron, J.-Y. Shan, D. Puggioni, N. Z. Koocher, K. Yamaura, Y. Shi, J. M. Rondinelli, and D. Hsieh, *Nature Communications* **10**, 3217 (2019).
- [44] P. W. Anderson and E. I. Blount, *Phys. Rev. Lett.* **14**, 217 (1965).
- [45] A. Ianculescu, Z. Mocanu, L. Curecheriu, L. Mitoseriu, L. Padurariu, and R. Truc, *Journal of Alloys and Compounds* **509**, 10040 (2011).
- [46] A. I. Ali and S. H. Kaytbay, *Materials Sciences and Application* **2**, 716 (2011).
- [47] L. W. Martin and D. G. Schlom, *Current Opinion in Solid State and Materials Science* **16**, 199 (2012).
- [48] D. G. Schlom, L.-Q. Chen, C. J. Fennie, V. Gopalan, D. A. Muller, X. Pan, R. Ramesh,

and R. Uecker, MRS Bulletin **39**, 118130 (2014).

REVIEWERS' COMMENTS

Reviewer #1 (Remarks to the Author):

NONE

Reviewer #2 (Remarks to the Author):

The authors have properly addressed my questions. I have no objection to the acceptance of the manuscript in current version.

Reviewer #3 (Remarks to the Author):

The manuscript has received very extensive reviews and comments. Overall, the interest from the reviewers has been quite strong. The authors have rather painstakingly responded to all comments/questions, adding a significant amount of results to their manuscript and supplement. They have certainly adequately responded to questions and comments from this reviewer, and I now can recommend the manuscript for publication.

O. Delaire

Reviewer #4 (Remarks to the Author):

The responses are satisfactory, I recommend to accept.